# Towards Neural Scaling Laws for Time Series Foundation Models

**Qingren Yao**[1,2], **Chao-Han Huck Yang**[3], **Renhe Jiang**[4], **Yuxuan Liang**[2*], **Ming Jin**[1*] **Shirui Pan**[1]

[1]Griffith University  [2]The Hong Kong University of Science and Technology (Guangzhou)
[3]NVIDIA Research  [4]The University of Tokyo

## Abstract

Scaling laws offer valuable insights into the design of time series foundation models (TSFMs). However, previous research has largely focused on the scaling laws of TSFMs for in-distribution (ID) data, leaving their out-of-distribution (OOD) scaling behavior and the influence of model architectures less explored. In this work, we examine two common TSFM architectures—encoder-only and decoder-only Transformers—and investigate their scaling behavior on both ID and OOD data. These models are trained and evaluated across varying parameter counts, compute budgets, and dataset sizes. Our experiments reveal that the negative log-likelihood of TSFMs exhibits similar scaling behavior in both OOD and ID settings. We further compare the scaling properties across different architectures, incorporating two state-of-the-art TSFMs as case studies, showing that model architecture plays a significant role in scaling. The encoder-only Transformers demonstrate better scalability than the decoder-only Transformers in ID data, while the architectural enhancements in the two advanced TSFMs primarily improve ID performance but reduce OOD scalability. While scaling up TSFMs is expected to drive performance breakthroughs, the lack of a comprehensive understanding of TSFM scaling laws has hindered the development of a robust framework to guide model scaling. We fill this gap in this work by synthesizing our findings and providing practical guidelines for designing and scaling larger TSFMs with enhanced model capabilities.

## 1 Introduction

Time series analysis is an important piece of data mining, facilitating decision-making and scientific inference across various domains (Zhang et al., 2023). As an important analysis task, time series forecasting has long been studied and drives a wide range of practical applications, from energy, climate and quantitative finance to urban computing and system management (Jin et al., 2023; Nie et al., 2024; Wen et al., 2024). Various methods have been proposed for this task, ranging from classical statistic models (Hyndman & Athanasopoulos, 2013), bespoke dynamical models (Prado, 2020), to the more recent deep-learning based approaches (Wen et al., 2022). Despite their competitive performance, the methods are typically designed for specific tasks, poor to generalize to other domains (Fan et al., 2023; Rasul et al., 2023). Concurrently, we are witnessing a paradigm shift in time series forecasting from task-specific models to universal models, with the emergence of time series foundation models (TSFMs). Timer (Liu et al., 2024), Moirai (Woo et al., 2024), and more recently proposed Time-MoE (Shi et al., 2024b) show trends of scaling in both data volume and model size, aiming to achieve performance breakthroughs through more resource investment.

The neural scaling law (Kaplan et al., 2020) quantitatively describes how model performance grows with the scaling of three basic training factors: model parameters, computational resources and training dataset size. Establishing such scaling laws is crucial for developing TSFMs, as it provides a framework for predicting expected performance gains, enabling the community to rationally allocate efforts toward key designs. The exploration on scaling laws for TSFMs is still in an initial stage; recent research has primarily focused on studying ID scaling behavior (Edwards et al.,

---

*Correspondence to: Y. Liang <yuxliang@outlook.com> and M. Jin <mingjinedu@gmail.com>.

2024; Shi et al., 2024a). In practical applications, TSFMs primarily face challenges from unseen scenarios (Wang et al., 2024), where OOD forecasting capability is most critical. This raises an unresolved question: *do neural scaling laws also apply to predict out-of-distribution forecasting performance?* Moreover, various architectures of TSFMs have been arising, but they typically focus on performance improvement at specific scales. No studies have investigated the scaling behaviors across different architectures, leaving a key question unanswered: *how do model architectures affect scalability?* Although we are seeing an increasing investment in training resources for TSFMs, the bottlenecks and potential driving factors for developing larger TSFMs remain unclear. This raises another practical question: *how to design TSFMs from the perspective of scalability?*

In this paper, we aim to provide empirical answers to the above research questions. To investigate the scaling laws in OOD scenarios, we trained a family of encoder-only Transformer-based TSFMs, varying three basic training factors: model sizes, compute budgets, and training set sizes. We evaluated their performance on both ID and OOD test sets and established scaling laws for three training factors in each scenario. To examine the impact of model architecture on scaling behavior, we trained decoder-only Transformer based TSFMs and compared them with the encoder-only versions. Additionally, we included two state-of-the-art TSFMs, Moirai (Woo et al., 2024) and Chronos (Ansari et al., 2024), as case studies for detailed analysis. Our experiment results suggest that the negative log-likelihood loss of TSFMs exhibits similar scaling behavior in both OOD and ID scenarios; encoder-only Transformers have a similar scalability with decoder-only Transformers, with a slight advantage in ID data; the architectural modifications introduced by two advanced TSFMs mainly improve ID performance but compromise OOD scalability. Based on the findings and comparative analysis, we finally provided design principles for TSFMs from a scaling perspective.

Our contributions are summarized as follows:

- **Scaling laws across data distributions.** We extend the scaling laws for TSFMs from ID scenarios to OOD scenarios across three core training factors: model size, computational resources, and dataset size, establishing a foundation for predicting expected OOD performance gains of TSFMs.
- **Scaling laws across model architectures.** We investigate the scaling patterns of different TSFM architectures, showing the scalability varies depending on the model architecture and the design of specific modules.
- **Scaling laws-guided design principles.** We provide practical design principles for TSFMs from the perspective of data, model and compute scaling, via analyzing the commonalities and differences in scaling behaviors across data distributions and model architectures

## 2 PRELIMINARY

To investigate the scaling laws of TSFMs, we curated a large, diverse, and balanced dataset for pre-training. Leveraging this dataset, we trained both *encoder-only* and *decoder-only* transformers within two state-of-the-art TSFMs: Moirai and Chronos. For comparative analysis, we evaluated these models on (i) *in-distribution* and (ii) *out-of-distribution* test sets, focusing on key performance metrics to examine the scaling behavior across architectures.

### 2.1 DATASETS

A large scale, diverse, balanced and high quality pre-training dataset is the foundation to build FMs. To this end, we constructed our time series corpus for TSFM pre-training from the large-scale open time series archive, LOTSA (Woo et al., 2024). The corpus comprises approximately 17B time points from 39 datasets spanning seven distinct domains. To ensure that the model performs fairly across all domains, we maintained a balanced ratio of data from different domains. Furthermore, we performed quality filtering on the corpus by constraining the signal-to-noise ratio of a time series to be greater than 20 dB, ensuring that the pre-training corpus exhibits strong predictability. A detailed breakdown of the data sources is provided in Appendix A, with a summary in Table 1.

To assess the impact of pre-training data scale on model performance, we partitioned the corpus into subsets containing 10M, 100M, and 1B time points, ensuring that each subset maintained similar diversity. For each subset, 95% of the data was allocated for model training, with the remaining 5% reserved as a validation set to evaluate in-distribution forecasting performance. Additionally,

Table 1: **Dataset summary.** M indicates million and B indicates billion.

| Domain | Transport | Climate | Energy | CloudOps | Health | Sales | Web | Total |
|---|---|---|---|---|---|---|---|---|
| Datasets | 8 | 2 | 14 | 3 | 9 | 1 | 2 | 39 |
| Time points | 4.82B | 4.73B | 4.76B | 2.15B | 232M | 140M | 40M | 16.8B |
| Proportion | 28.52% | 28.06% | 28.21% | 12.76% | 1.38% | 0.83% | 0.24 % | 100% |

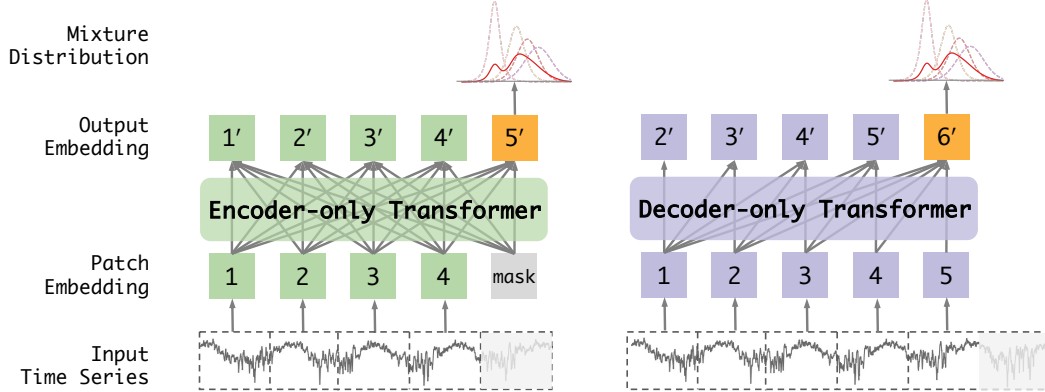

Figure 1: **Architectures of Baseline Time Series Foundation Models.** As the most widely used two Transformer architectures, encoder-only Transformer and decoder-only Transformer are selected to as our baseline. A time series is divided into multiple patches, each treated as a token and fed into the Transformer model. The shaded patches represent the future horizon to be predicted.

we used a subset from a widely recognized long-sequence prediction benchmark (Wu et al., 2023) to test the model's *out-of-distribution* forecasting capabilities. To further enhance the reliability, we also incorporated a subset of the Monash dataset (Godahewa et al., 2021) as additional OOD test data. The details of the dataset composition and properties are provided in Appendix A Table 3.

## 2.2 MODELS

TSFMs are predominantly built upon the Transformer architecture (Wen et al., 2022). For our baseline models, we selected two widely adopted architectures: the encoder-only Transformers (Woo et al., 2024) (Moirai) and the decoder-only Transformers (Ansari et al., 2024) (Chronos). The primary distinction between them lies in the attention mechanisms applied to the inputs, as illustrated in Figure 1. To better adapt them for time series forecasting, we introduce three key modifications in input layer, positional encoding and prediction head. More details are given in Appendix B.

**Patch Embedding.** There are several approaches for generating inputs for transformer-based TSFMs, including point embedding, patch embedding, and lagged feature embedding. Due to the high computational cost of point embedding for long sequences and the limited robustness of lagged feature embedding, we adopt patch embedding in our models. This method, initially introduced by Vision Transformers (Dosovitskiy et al., 2020) and later adapted by PatchTST (Nie et al., 2023) for time series forecasting, divides the time series into non-overlapping segments, which are then projected into a feature space.

**Rotary Position Embedding.** This technique (RoPE) has rapidly gained popularity as a positional encoding method in recent large language models (Su et al., 2024). Given the improved pre-training efficiency observed with RoPE (Woo et al., 2023), we adopt RoPE as a replacement for the original Transformer's positional encoding. RoPE encodes absolute positions using a rotation matrix while embedding relative position dependencies directly into the self-attention mechanism.

**Mixture of Distributions.** Our models are designed to predict the probability distribution of future time series. However, real-world time series often exhibit complex distributions, including outliers, heavy tails, and extreme skew, which pose significant challenges for accurate modeling. To address

these complexities, we incorporate a more flexible output likelihood by utilizing Student-T mixture models (Flunkert et al., 2017). Compared to the commonly used Gaussian mixture models, Student-T mixture models offer greater robustness in handling outliers and heavy-tailed distributions. An empirical comparison between the two mixture distributions is shown in Appendidx B Figure 9.

Our models are characterized by several key hyper-parameters: the number of layers ($n_{\text{layer}}$), the input/output dimensions of the residual stream ($d_{\text{m}}$), the dimensions of the intermediate feed-forward layers ($d_{\text{ff}}$), the number of attention heads per layer ($n_{\text{heads}}$), and the dimension of the attention output ($d_{\text{head}}$). The overall model size can be expressed as:

$$\begin{aligned} N &\approx n_{\text{layer}} \left(4d_{\text{m}}n_{\text{heads}}d_{\text{head}} + 2d_{\text{m}} * d_{\text{ff}}\right) \\ &= 2d_{\text{m}}n_{\text{layer}} \left(2n_{\text{heads}}d_{\text{head}} + d_{\text{ff}}\right) \\ &= 12n_{\text{layer}}d_{\text{m}}^2 \quad \text{with the standard} \quad n_{\text{head}} \cdot d_{\text{head}} = d_{\text{m}} = d_{\text{ff}}/4, \end{aligned} \tag{1}$$

where the embedding layer, prediction head, biases and other sub-leading terms are excluded for a cleaner scaling laws. The embedding layer uses a patch size of 32 with $32d_{\text{m}}$ parameters. The mixture distribution prediction head comprises multiple independent linear layers that predict each Student-t mixture distribution parameter for a patch separately, with $512d_{\text{m}}$ parameters in total. In the study, we explore models with $\sim 10^3$ to $\sim 10^8$ trainable parameters.

## 2.3 TRAINING AND EVALUATION DETAILS

In this study, we focus exclusively on uni-variate time series forecasting to avoid the confounding effects introduced by multivariate time series, such as variable interactions, correlations, and the complexities of modeling multivariate relationships. Future research will address these factors, aiming to establish more comprehensive scaling laws for multivariate time series models.

**Training Details**. Our training objective is to optimize the mixture distribution log-likelihood. We utilize the AdamW optimizer with a batch size of 128, and a maximum learning rate of $10^{-3}$ with a linear warm-up of $10^4$ training steps, followed by cosine decay for the remaining $9 \times 10^4$ steps. To facilitate learning data representations across diverse domains with varying series lengths and sample sizes, we visited each sample with probability $p_i = t_i/T$, where $t_i$ is the series' time points and $T$ is the corpus' total time points. In addition, we follow the approach used in Moirai (Woo et al., 2024) and Timer (Liu et al., 2024) by capping the sampling probability at 0.05 to ensure a more balanced contribution from each dataset. We then randomly selected a segment from each chosen sample.

**Evaluation Details**. We evaluate the model on a randomly selected 10% subset of the test data every $10^3$ steps to reduce computational costs. For performance measurement, we observed that non-normalized metrics like MAE and MSE are highly sensitive to the amplitude of time series data, often causing the overall average to be disproportionately influenced by high-amplitude datasets. To mitigate this issue, we primarily use the normalized metric, mean absolute percentage error (MAPE), along with the negative log-likelihood (NLL), to assess forecasting performance. For a more comprehensive understanding of TSFM scaling laws, we also include additional results using symmetric mean absolute percentage error (SMAPE), mean absolute scaled error (MASE), and continuous ranked probability score (CRPS) in Appendix D.4. Detailed descriptions of these metrics are provided in the Appendix C.3.

## 3 SCALING LAWS FOR TIME SERIES FOUNDATION MODELS

In this section, we first present experimental results using the encoder-only Transformer to explore scaling laws across different data distributions. Following this, we conduct a comparative study on the scaling behavior of encoder-only and decoder-only TSFMs, Chronos and Moirai, to investigate how various scaling factors influence the characteristics of time series models.

### 3.1 SCALING LAWS ACROSS DATA DISTRIBUTIONS

**Parameter Scaling.** In Figure 2, we display the ID and OOD performance of a wide variety of encoder-only Transformers, ranging from small models with 1K parameters through large models

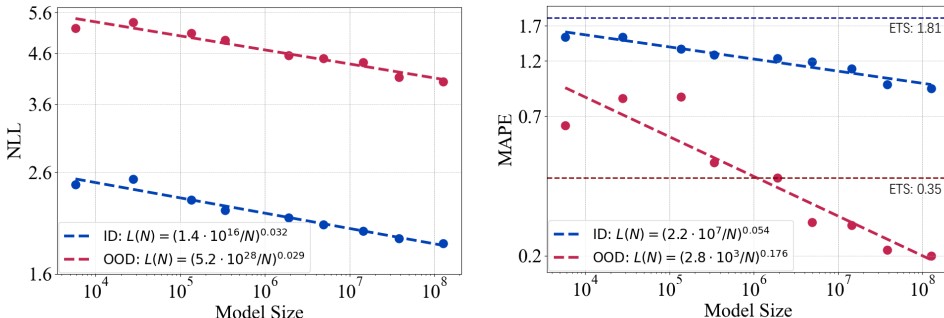

Figure 2: **Parameter Scaling.** The scaling effect of total trainable model parameters on the in-distribution (ID) and out-of-distribution (OOD) forecasting performance, which is evaluated using NLL and MAPE metrics. When evaluated with NLL, both ID and OOD results follow an approximate power law scaling with parameter count, exhibiting consistent trends across different data distributions. The blue and red horizontal dashed lines represent the baselines of the exponential smoothing (ETS) forecasting method.

with 100M parameters. We trained models on the full pre-training corpus to convergence and report the minimum NLL and MAPE. We can see that both ID and OOD performance roughly follow power-law behavior over five orders of magnitude in model sizes. Formally, the power law can be expressed as:

$$L(N) \approx \left(\frac{N_c}{N}\right)^{\alpha_N},\tag{2}$$

where $L$ is the performance metric function (i.e., MAPE, or NLL), $N$ is a given parameter count, $N_c$ is the normalization coefficient, and $\alpha_N$ is the exponent value that indicates the degree of performance improvement expected as we scale up $N$.

Observing the NLL metric, the lines fitting the scaling laws for both ID and OOD data exhibits a roughly constant shift and close slopes. This implies that while models incur a consistent performance bias when transferred to OOD data, their scaling patterns correlate well with their performance on the ID data. When evaluated using MAPE, the power-law for OOD scenario shows a bigger exponent value than ID scenario. This indicates that increasing model size yields greater improvements in OOD performance than ID performance. In other words, for models with weak OOD generalization capabilities, increasing model size may enable them to perform equally well on both ID and OOD data.

To evaluate whether the benefits of large-scale pre-training are warranted, we compare the pre-trained models with the classical exponential smoothing (ETS) forecasting method. The results indicate that the pre-trained models consistently outperform ETS on ID data and progressively excel on OOD data as the model size increases. This suggests that pre-trained models must reach a certain scale, at least 3M parameters in this case, to demonstrate a level of superiority on OOD data that justifies their high pre-training cost.

**Compute Scaling.** Following the similar method in (Kaplan et al., 2020), we estimate the compute budget using the formula $C = 6NBS$, where $B$ is the batch size, $S$ is the number of parameter updates, i.e. the input sequence length, and 6 is the factor to account for the forward and backward passes. The ID and OOD test loss for compute budget varying over six orders of magnitude are shown in Figure 3. We see that the optimal results for each compute budget are achieved by different model sizes $N$, but the lowest loss decreases according to a approximate power law with respect to the amount of training compute. The lowest losses appear as the heavy lines, which can be fit with

$$L(C) \approx \left(\frac{C_c}{C}\right)^{\alpha_C}.\tag{3}$$

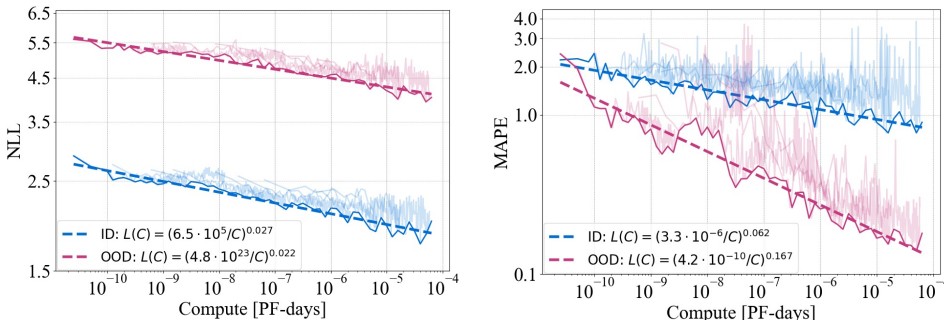

Figure 3: **Compute Scaling.** The computation scaling results indicate that model performance scales approximately according to a power law with increasing compute, consistent across both ID and OOD scenarios. The ID and OOD results illustrate that there is an lower bound for loss and MAPE on both test data under a given computational budget.

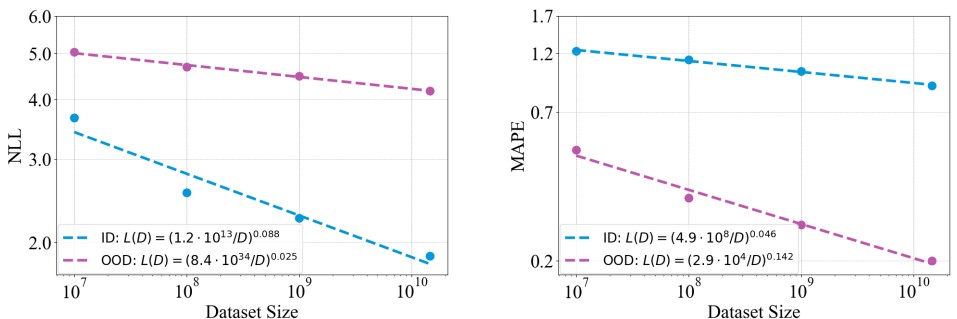

Figure 4: **Data Scaling.** The blue and red plots illustrate how data volume affects the ID and OOD forecasting performance of encoder-only Transformers, evaluated using NLL and MAPE metrics. The results indicate that in both scenarios, model performance scales approximately as a power law with data volume.

We observed significant noise in NLL and MAPE during training, which may be caused by the learning rate scheduler and the random sampling evaluation strategy. Similar to parameter scaling, the ID and OOD NLL show similar scaling patterns; however, when evaluated using MAPE, the OOD power law shows larger exponent values than the ID power law.

**Data Scaling.** We display empirical trends for the performance as function of dataset size $D$ in Figure 4. For the trend, we trained multiple 1B encoder-only Transformers on a series of subsets of the pre-training dataset and report the averaged evaluation results during training. We see that the NLL and MAPE can be fit well with simple power-law

$$L(D) \approx \left( \frac{D_c}{D} \right)^{\alpha_D}. \tag{4}$$

Different from the parameter scaling and computational scaling, we found that when evaluated using NLL, ID and OOD performance do not exhibit the same scaling behavior. Instead, ID performance is more sensitive to the scaling of dataset size compared to OOD performance. However, similar to the observations on the other two factors, the impact of data scaling on MAPE in OOD data is greater than its impact on MAPE in ID data. This suggests that the scaling of various factors yields greater improvements in OOD performance than ID performance.

**Cross-distribution Scaling Effects.** We summarize the key findings on how model ID and OOD performance scales with the model parameters, data volume, and compute. (1) In both ID and OOD uni-variate time series forecasting, model performance follows a simple power law as a function of model parameters, data volume, and compute. (2) For NLL, the model exhibits similar scaling

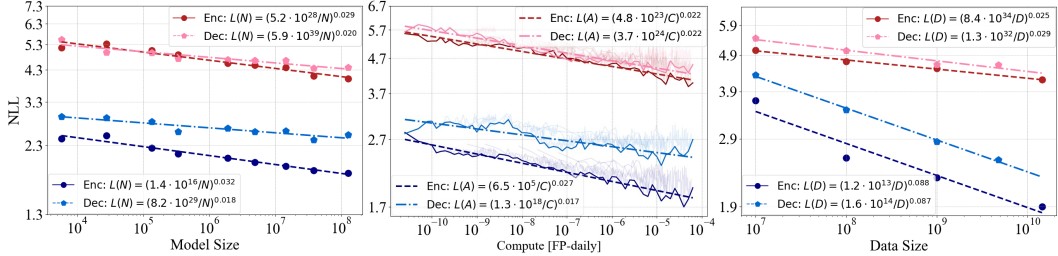

Figure 5: **Scaling Laws of Encoder-only vs. Decoder-only Transformer.** This figure presents a comparison of scaling behaviors on NLL between encoder-only and decoder-only Transformer across three different axes: number of parameters, compute, and dataset size. Overall, both models exhibit similar scalability patterns with respect to model parameters, computation and dataset sizes across ID and OOD data, but differ in ID performance.

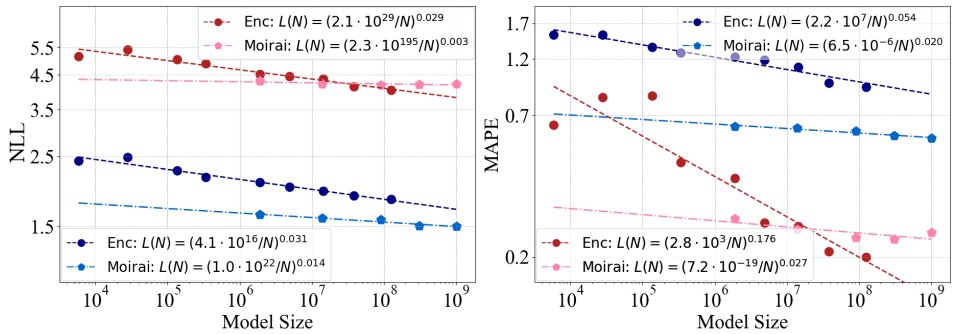

Figure 6: **Scaling Laws of Encoder-only Transformer vs. Moirai.** This figure compares parameter scaling between the encoder-only Transformer and Moirai. While Moirai demonstrates notable improvements on ID test data compared to the baseline, certain design choices may hinder its scalability on OOD data.

patterns in both ID and OOD scenarios, in terms of model size or compute resources. (3) When using MAPE as the metric, scaling of all three factors results in greater improvements in OOD performance compared to ID performance.

## 3.2 SCALING LAWS ACROSS MODEL ARCHITECTURES

The above results suggest that the power-law captures the scaling behavior of encoder-only Transformers in both ID and OOD scenarios. Similarly, we analyze the scaling properties of decoder-only Transformers, along with two other state-of-the-art TSFMs, Chronos and Moirai, to assess the impact of model architectures on scaling behavior. These models are trained on the dataset we built with the same training setup.

**Encoder-only vs. Decoder-only Transformer.** Figure 5 presents a comparison of the scaling patterns between encoder-only and decoder-only Transformers. In parameter scaling, the encoder-only Transformer shows a slight advantage, with a marginally higher power-law exponent on ID data and lower NLL values. For compute and data scaling, the two architectures demonstrate nearly identical scalability across ID and OOD settings, with some performance differences observed on ID data. Overall, the two architectures exhibit similar scalability, aside from a performance difference observed in ID data.

**Encoder-only Transformer vs. Moirai.** Moirai is a TSFM based on an encoder-only Transformer architecture. It introduces "any-variate" attention to capture relationships between multiple variables. Additionally, Moirai incorporates a multi-scale patch embedding to handle different frequency patterns, and a more diverse mixture distribution to model real-world probability distributions. Figure 6 show a comparison between the scaling behavior of encoder-only Transformer and

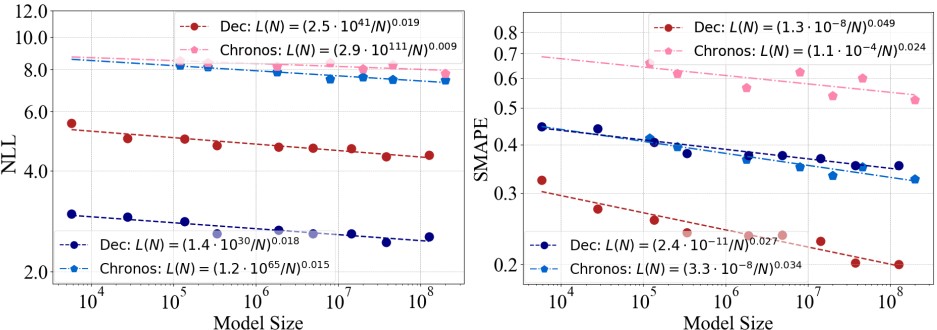

Figure 7: **Scaling Laws of Decoder-only Transformer vs. Chronos.** This figure presents a comparison of parameter scaling between decoder-only Transformer and Chronos. The design introduced by Chronos enhances ID time series forecasting, but fails to improve the generalization of OOD.

**Moirai.** On ID data, Moirai demonstrates better performance. However, for OOD data, as the number of model parameters increases, Moirai is gradually surpassed by the encoder-only Transformer. Comparing the power-law lines of the two models, we see that Moirai shows a smaller slope, indicating relatively weaker scalability. Collectively, Moirai shows significant improvements on ID time series forecast than our baseline, but some designs may limit its scalability on OOD data.

**Decoder-only Transformer vs. Chronos.** Chronos-T5 is an encoder-decoder Transformer-based TSFM that, like decoder-only Transformers, follows an auto-regressive prediction approach but uses a separate encoder to extract contextual information. It adopts point-wise prediction and transform numerical regression into discrete probability prediction. Figure 7 compares the scaling patterns of decoder-only Transformers and Chronos-T5. When evaluated with NLL, Chronos-T5 exhibits power laws with very small exponents, indicating limited scalability. We attribute this to the discrete probability prediction, as NLL on a discrete distribution is not distance-sensitive, meaning the loss remains high unless the predicted value exactly matches the label. Compared to NLL, symmetric mean absolute percentage error (SMAPE, definition is shown in Appendix C.3) is a more appropriate metric for evaluating the two models. Chronos-T5 shows a slight advantage in performance and scalability on ID data. However, on OOD data, the decoder-only Transformer performs better. Despite some performance differences, both models exhibit similar scaling patterns on OOD data. Overall, the design improvements in Chronos-T5 enhance ID time series forecasting, but they do not effectively improve OOD generalization.

**Scaling Effects Summary.** Our analysis of the scaling properties across architectures reveals several key findings: (1) Model architecture and design play a crucial role in determining scalability. (2) Encoder-only models and decoder-only models show a similar scalability on OOD data; while encoder-only models have a slightly higher parameter scalability and a certain performance advantage on ID data. (3) While Chronos and Moirai improve ID forecasting, these gains do **not** extend effectively to OOD scenarios.

# 4 DESIGN PRINCIPLES FOR TIME SERIES FOUNDATION MODELS

Building on our findings regarding the scaling laws in TSFMs, we elaborate design principles to guide the development of scalable models. These principles are framed around three key dimensions: training data, model parameters and architecture, as well as computational resources.

**Training Data.** Our experiments show that increasing the size of the training dataset leads to a greater performance improvement on OOD data compared to ID data. Enlarging the pre-training dataset is crucial for achieving better generalization. Based on Equation 4, we can expect that doubling the size of the pre-training dataset will result in an OOD MAPE reduction to approximately 90% of its previous value, and an ID MAPE reduction to approximately 97%. But maintaining diversity within the dataset is equally important while increasing the data volume.

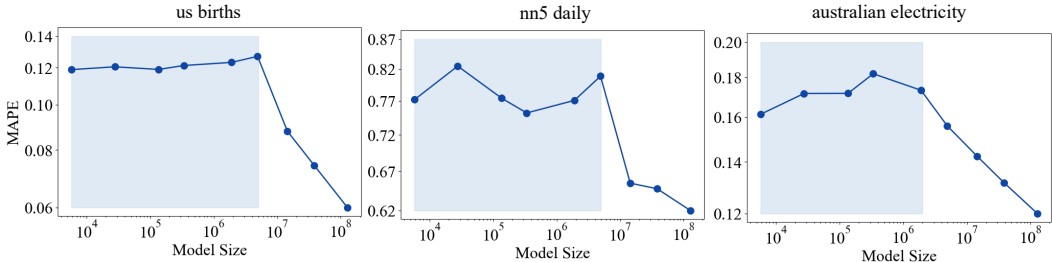

Figure 8: **Case studies on the "Emergent Abilities" of scaling time series models.** We present three examples of zero-shot (**unseen**) out-of-distribution (OOD) time series prediction. In the us births, nn5 daily, and australian electricity datasets, we observed that model behavior deviates from expected power law patterns, instead exhibiting characteristics more akin to *emergent phenomena*.

**Model Parameters and Architecture.** Our study highlights that model size is the most critical factor for improving TSFM performance. Compared to collecting more data, increasing model size yields the greater benefit for both ID and OOD forecast. However, as model size scales up, data bottlenecks inevitably emerge. From the ratio of exponents in Equations 2 and 4, we infer that scaling up model size should be accompanied by a sublinear increase in dataset size. Specifically, this relationship can be approximated as $D \propto N^{\frac{\alpha_N}{\alpha_D}} \sim N^{0.8}$, meaning that doubling the model size requires roughly 1.7 times more data. In terms of model architecture, we found some modifications facilitate ID performance but can not generalize well to OOD scenarios. Model design can impact scalability, and a good design should promote performance, generalization, and scalability.

**Compute.** The power law we established indicates that there is an unbreakable lower bound for NLL loss and MAPE under a given computational budget. This means that, with other factors remaining constant, as the model size increases, more compute resources must be invested to achieve better performance. Our findings suggest that compute resources should prioritize scaling up model size rather than extending training time for fixed models as larger models are more sample-efficient than smaller models. Moreover, different training objectives or model architectures can significantly affect this performance bound, exploring a compute-efficient training paradigm will be promising.

## 5 RELATED WORKS

**Neural Scaling Laws**. Neural scaling laws seek to provide a predictive framework for optimizing the allocation of computational resources to maximize model performance. In language domains, Kaplan et al. (2020) demonstrated that performance follows a power-law relationship, improving as more computational resources, parameters, and data are utilized. Subsequent research has expanded this to predict other factors, such as downstream task performance (Isik et al., 2024) and inference time (Sardana et al., 2024). In vision domains, scaling laws have been explored in areas like discriminative modeling (Hestness et al., 2017) and visual auto-regressive modeling (Henighan et al., 2020). Recently, Edwards et al. (2024) introduced scaling laws for large time series models, showing that performance scales according to a power law with model size, compute, and dataset size. Shi et al. (2024a) examined the effect of time-series forecasting horizon on model scaling behavior, offering a theoretical framework to explain its influence. However, both studies have focused on in-distribution scenarios, leaving the investigation of scaling laws in out-of-distribution contexts largely unexplored.

**Time Series Foundation Models.** Foundation models (Das et al., 2024; Goswami et al., 2024) represent a new paradigm aimed at generalizing across diverse domains and tasks by leveraging knowledge from large-scale data in the pre-training phase. They have significantly advanced time series forecasting, particularly in zero-shot scenarios, where predictions are made on data from previously unseen domains. For instance, Woo et al. (2024) introduced Moirai, an encoder-only transformer architecture that employs an "any-variate" attention mechanism to capture dependencies in multivariate time series. Ansari et al. (2024) proposed a method that tokenizes time series values through scaling and quantization into a fixed vocabulary, training a series of transformer-based

models known as Chronos. Liu et al. (2024) developed Timer, a simple decoder-only transformer architecture designed for univariate time series forecasting, while Rasul et al. (2023) introduced Lag-Llama, a decoder-only transformer that integrates lags as covariates to improve forecasting accuracy. These models incorporate various modifications to the standard Transformer architecture for time series data. However, the impact of these changes on model scaling properties has not been systematically studied. As model size increases, it remains an open question whether these modifications will continue to enhance performance.

# 6 DISCUSSION

## 6.1 ADDITIONAL STUDIES

We also conduct the following analysis to better understand the scaling behaviors of TSFMs. Due to the space limit, see their details in Appendix D.2 to D.4 .

**Emergent Behaviors.** Figure 8 shows some examples of zero-shot OOD time series prediction, where the model's performance remains low until the model size reaches a critical threshold, after which performance improves significantly. This scaling behavior deviates from the previously observed power law and is more akin to emergent phenomena (Wei et al., 2022).

**Sample Efficiency.** Appendix D.2, Figure 14 shows the evaluation results during training. Large models are more sample-efficient than small models, reaching better performance with the same optimization steps and using fewer time points.

**Scaling Pattern Depends on Data Distributions.** Appendix D.3, Figure 15 - 19 show the scaling behavior of TSFMs on the Monash subset, suggesting that OOD scaling behaviors varies depending on the relationship between the unseen data distribution and the pre-training data distribution.

**Scaling Pattern Depends on Performance Metrics.** Appendix D.4, Figure 15 - 19 provides a comprehensive analysis of scaling behavior across five performance metrics: NLL, MAPE, SMAPE, MASE, and CRPS. All metrics exhibit a consistent decreasing trend, approximately following a power law.

## 6.2 CONCLUSION

We have observed consistent scaling of encoder-only Transformer NLL with parameter count, training computation and dataset size on both ID and OOD test data, as encapsulated in a power law. The experimental results show that as the number of parameters, computational resources, and training data increase, both models' ID and OOD performance will continue to improve. Furthermore, our established laws suggest that for ID and OOD forecasting performance, larger model may be more important than more data.

We have also study the scaling properties of two common types of TSFMs, encoder-only Transformer and decoder-only Transformer. The two architectures show similar scalability in OOD data across model size, computational resources, and training data. While encoder-only models have a slightly better parameter scalability and performance in ID data based our evaluation setting. Moreover, we further include two advanced TSFMs, Chronos and Moirai, as specific cases to study the impact of module design on scaling behavior. We find that both Chronos and Moirai improve ID forecasting performance, but these gains can not translate effectively to OOD scenarios. Overall, model architecture and module design is important influence factor for scalability.

Our experiments independently investigated the scaling behavior of TSFMs in terms of parameter count, computational resources, and training data size, assuming other factors were unlimited. In future work, it would be valuable to develop a unified model to explore the relationships between these factors. This could help guide the community in optimizing resource allocation when one resource is limited. Additionally, multiple studies have shown that model performance varies across different context window lengths and forecast horizons. In future research, we plan to incorporate the impact of context length and forecast horizon on scalability.

**Acknowledgments and Disclosure of Funding**

This research was partly supported by Australian Research Council (ARC) under grants FT210100097 and DP240101547 and the CSIRO – National Science Foundation (US) AI Research Collaboration Program. This work has been also supported by NVIDIA Academic Grant from the NVIDIA Developer Program 2025.

**Ethics Statement**

Our research is dedicated exclusively to addressing scientific challenges and does not involve human participants, animals, or materials that pose environmental concerns. We anticipate no ethical risks or conflicts of interest.

**Reproducibility Statement**

We provide the implementation details in Appendices B and C, including model configurations and training & evaluation details. The source code and related source of this work are available at `https://github.com/Qingrenn/TSFM-ScalingLaws` for reproducibility.

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

# A  PRE-TRAINING DATASETS

Previous work has proposed several public large-scale time series datasets, e.g. LOTSA (Woo et al., 2024) and UTSD (Liu et al., 2024). Instead of using these datasets directly, we built a pre-training corpus through data filtering. There are three main reasons for this: First, these datasets contain numerous periodic time series with substantial pattern redundancy. Second, these datasets show a heavily long-tailed distribution, with multiple domains accounting for less than 5% of the total. Third, data quality is variable, with some time series missing many observations or having low signal-to-noise ratios. To address these issues, we developed a data filtering pipeline:

**Deduplication.** In these public datasets, we observed significant redundancy across many subsets. This redundancy typically manifests as numerous repeated patterns within individual samples or high similarity between multiple samples. Such redundancy can negatively impact training efficiency. To address this issue, we applied a down-sampling strategy for datasets with redundant samples based on the data sampling period.

**Quality Filtering.** We focused on selecting time series data that have no missing values and a signal-to-noise ratio (SNR) greater than 20 dB to ensure better predictability. To calculate the SNR, we first used Fourier transform to identify the low-frequency components of the time series. We then applied a low-pass Butterworth filter to extract these low-frequency components as the signal, with the remaining residuals considered as noise.

**Domain Balancing.** In the LOTSA dataset, the climate domain accounts for nearly 90% of the data, while the remaining seven domains collectively represent only 10%. To create a more balanced pre-training dataset, we selected the transport domain, which contains 4.8 billion time points, as the reference. We then scaled the data from the other domains to a similar magnitude.

Following the aforementioned processing steps, we compiled a high-quality time series dataset containing a total of 16.8 billion time points across 7 domains. The data volume is sufficient to support training models with parameters ranging from $10^3$ to $10^8$ parameters. Furthermore, most domains include at least 100 million time points, ensuring that even when the dataset is divided into subsets of various sizes, domain diversity is preserved. Below, we introduce the datasets for each domain and outline the key properties of the datasets after processing, including domain type, sampling frequency, number of time series, total number of observations, and data source.

**Climate.** The climate data are sourced primarily from the ERA5 and CMIP6 datasets, which provide time series for various climate-related variables, including temperature, humidity, and pressure levels. During data curation, we observed a high degree of similarity across yearly data. To reduce redundancy, only two years of data from ERA5 and CMIP6 were included.

**Energy.** The energy data primarily come from the BuildingsBench dataset, which provides time series data on residential and commercial building energy consumption. After applying quality filtering based on SNR, we selected samples from the Buildings-900K, BDG-2, and Sceaux datasets. Additionally, high-quality data from the ProEnFo library, including the GEF, ELF, PDB, Spanish, and Covid19_Energy datasets, were incorporated.

**Transport.** The primary source for transport data is the LargeST traffic dataset, covering traffic flow in California from 2017 to 2021. We also integrated datasets from LibCity, PEMS, Loop Seattle, and Q-Traffic into our corpus.

**Cloud Operations.** The cloud operations data is from the large-scale CloudOps time series datasets, which measures various variables such as CPU and memory utilization.

**Healthcare.** We include a diverse sets of healthcare data from the ucr time series archive.

**Web.** The web data comprises the Kaggle Web Traffic Weekly dataset and the Wiki-Rolling dataset.

**Sales.** Here we use the Favorite Sales dataset.

Table 2: Pre-training datasets and key properties.

| Dataset | Domain | Frequency | # Time Series | # Obs. | Source |
|---|---|---|---|---|---|
| CMIP6 | Climate | 6H | 196,608 | 2,870,476,800 | Nguyen et al. (2023) |
| ERA5 | Climate | H | 212,992 | 1,860,698,112 | Nguyen et al. (2023) |
| Buildings900K | Energy | H | 538,577 | 4,718,473,097 | Emami et al. (2023) |
| Australian Electricity | Energy | 30T | 5 | 1,153,584 | Godahewa et al. (2021) |
| BDG-2 Bear | Energy | H | 91 | 1,482,312 | Emami et al. (2023) |
| BDG-2 Fox | Energy | H | 135 | 2,324,568 | Emami et al. (2023) |
| BDG-2 Panther | Energy | H | 105 | 919,800 | Emami et al. (2023) |
| Sceaux | Energy | H | 1 | 34,223 | Emami et al. (2023) |
| Solar Power | Energy | 4S | 1 | 7,397,222 | Godahewa et al. (2021) |
| Covid19 Energy | Energy | H | 1 | 31,912 | Wang et al. (2023b) |
| Spanish | Energy | H | 1 | 35,064 | Wang et al. (2023b) |
| Elecdemand | Energy | 30T | 1 | 17,520 | Godahewa et al. (2021) |
| PDB | Energy | H | 1 | 17,520 | Wang et al. (2023b) |
| GEF17 | Energy | H | 8 | 140,352 | Wang et al. (2023b) |
| GEF14 | Energy | H | 1 | 17,520 | Wang et al. (2023b) |
| ELF | Energy | H | 1 | 21,792 | Wang et al. (2023b) |
| Traffic Weekly | Transport | W | 862 | 82,752 | Godahewa et al. (2021) |
| Q-Traffic | Transport | 15T | 45,148 | 264,386,688 | Jiang et al. (2023) |
| PEMS04 | Transport | 5T | 921 | 15,649,632 | Jiang et al. (2023) |
| PEMS07 | Transport | 5T | 883 | 24,921,792 | Jiang et al. (2023) |
| PEMS08 | Transport | 5T | 510 | 9,106,560 | Jiang et al. (2023) |
| PEMS Bay | Transport | 5T | 325 | 16,941,600 | Jiang et al. (2023) |
| Loop Seattle | Transport | 5T | 1,809 | 33,953,760 | Jiang et al. (2023) |
| LargeST | Transport | 5T | 42,333 | 4,452,510,528 | Liu et al. (2023) |
| Azure VM Traces 2017 | CloudOTS | 5T | 159,472 | 885,522,908 | Woo et al. (2023) |
| Borg Cluster Data 2011 | CloudOTS | 5T | 286,772 | 1,075,105,708 | Woo et al. (2023) |
| Alibaba Cluster Trace 2018 | CloudOTS | 5T | 116,818 | 190,385,060 | Woo et al. (2023) |
| Wiki-Rolling | Web | D | 47,675 | 40,619,100 | Alexandrov et al. (2020) |
| Kaggle Web Traffic Weekly | Web | W | 145,063 | 16,537,182 | Godahewa et al. (2021) |
| Favorita Sales | Sales | D | 111,840 | 139,111,860 | Woo et al. (2024) |
| PigArtPressure | Health | - | 312 | 624,000 | Dau et al. (2018) |
| SelfRegulationSCP1 | Health | 0.004 SEC | 3,366 | 3,015,936 | Dau et al. (2018) |
| SelfRegulationSCP2 | Health | 0.004 SEC | 2,660 | 3,064,320 | Dau et al. (2018) |
| TDBrain | Health | 0.002 SEC | 28644 | 73,299,996 | Wang et al. (2023a) |
| MotorImagery | Health | 0.001 SEC | 24,192 | 72,576,000 | Dau et al. (2018) |
| PigCVP | Health | - | 312 | 624,000 | Dau et al. (2018) |
| AtrialFibrillation | Health | 0.008 SEC | 60 | 38,400 | Dau et al. (2018) |
| IEEEPPG | Health | 0.008 SEC | 15,480 | 15,480,000 | Tan et al. (2021) |
| BIDMC32HR | Health | - | 15,898 | 63,592,000 | Tan et al. (2021) |

## B  TIME SERIES MODELS

We define time series forecasting as the following problem: given a collection of multivariate time series samples with look back window $L : (\mathbf{x}_1, \ldots, \mathbf{x}_L)$ where each $\mathbf{x}_t$ at time step $t$ is a vector of dimension of $M$, our goal is to forecast $T$ future values $(\mathbf{x}_{L+1}, \ldots, \mathbf{x}_{L+T})$.

**Patch Embedding.** We split the input $(\mathbf{x}_1, \ldots, \mathbf{x}_L)$ into $M$ uni-variate time series $\mathbf{x}^{(i)} \in \mathbb{R}^{1 \times L}$, independently forecasting future time series for each variate. Each uni-variate time series $\mathbf{x}^{(i)}$ is first divided into non-overlapped patches. Specifically, given the patch length as $P$, the patching process will generate the a sequence of patches $\mathbf{x}_p^{(i)} \in \mathbb{R}^{P \times N}$ where $N = \left[ \frac{L}{P} \right]$ is the number of patches. Then the patches are mapped to the latent space of $d_m$ via a learnable linear projection $\mathbf{W}_p \in \mathbb{R}^{d_m \times P}$. In our baseline models, the patch size $P$ is set to 32.

**Rotary Position Embedding.** Rotary Positional Embedding (RoPE) is a type of position encoding that encodes absolute positional information with a rotation matrix and naturally incorporates explicit relative position dependency in self-attention formulation. In detail, RoPE incorporates absolute position information to the embedding and transform them into queries, keys through function:

$$ f_{q,k}(\mathbf{x}_m, m) = \mathbf{R}_{\Theta,m}^d \mathbf{W}_{q,k} \mathbf{x}_m $$

where

$$ \mathbf{R}_{\Theta,m}^d = \begin{bmatrix} \mathbf{M}_1 & & & \\ & \mathbf{M}_2 & & \\ & & \ddots & \\ & & & \mathbf{M}_{d/2} \end{bmatrix}, \mathbf{M}_j = \begin{pmatrix} \cos m\theta_j & -\sin m\theta_j \\ \sin m\theta_j & \cos m\theta_j \end{pmatrix} $$

is the rotary matrix with pre-defined parameters $\Theta = \{\theta_i = 10000^{-2(i-1)/d}, i \in [1, 2, \ldots, d/2]\}$, $\mathbf{W}_{q,k}$ is the learned query or key projection weights, and $\mathbf{x}_m \in \mathbb{R}^d$ is the embedding of the $m$ token. Next, applying RoPE to the dot product of query and key, we can obtain:

$$ \mathbf{q}_m^\top \mathbf{k}_n = \left( \mathbf{R}_{\Theta,m}^d \mathbf{W}_q \mathbf{x}_m \right)^\top \left( \mathbf{R}_{\Theta,n}^d \mathbf{W}_k \mathbf{x}_n \right) = \mathbf{x}_m^\top \mathbf{W}_q \mathbf{R}_{\Theta,n-m}^d \mathbf{W}_k \mathbf{x}_n $$

where $\mathbf{R}_{\Theta,n-m}^d = \left( \mathbf{R}_{\Theta,m}^d \right)^\top \mathbf{R}_{\Theta,n}^d$. In this way, RoPE naturally incorporates relative position information through rotation matrix product. In general, the self-attention enhanced with RoPE can be written as

$$ \text{Attention}(\mathbf{Q}, \mathbf{K}, \mathbf{V})_m = \frac{\sum_{n=1}^N \left( \mathbf{R}_{\Theta,m}^d \phi(\mathbf{q}_m) \right)^\top \left( \mathbf{R}_{\Theta,n}^d \varphi(\mathbf{k}_n) \right) \mathbf{v}_n}{\sum_{n=1}^N \phi(\mathbf{q}_m)^\top \varphi(\mathbf{k}_n)}. $$

where $\phi(\cdot), \varphi(\cdot)$ are usually non-negative functions, e.g. elu$(\cdot)$+1, or exp$(\cdot)$. Overall, unlike the traditional approach of adding positional information before the query and key projection, RoPE utilizes a rotary matrix to transform query and key embeddings, effectively leveraging the geometric properties of vectors.

**Mixture of Distributions.** As described in (Flunkert et al., 2017), our model predicts the parameters of a probability distribution, specifically a mixture of Student-t distributions. The probability density function for a random variable $x$ following a Student-t distribution is given by:

$$ p(x; \nu, \mu, \tau) = \frac{\Gamma\left(\frac{\nu+1}{2}\right)}{\Gamma\left(\frac{\nu}{2}\right)\sqrt{\pi\nu}\tau} \left( 1 + \frac{1}{\nu} \left( \frac{x - \mu}{\tau} \right)^2 \right)^{-\frac{\nu+1}{2}} $$

with parameters $\nu > 0, \mu \in \mathbb{R}, \tau > 0$ represent the degrees of freedom (df), location, and scale coefficient, respectively, and $\Gamma$ is the gamma function. We modeled the predicted distribution using a mixture of four Student-t distributions. Our baseline model employs independent linear layers

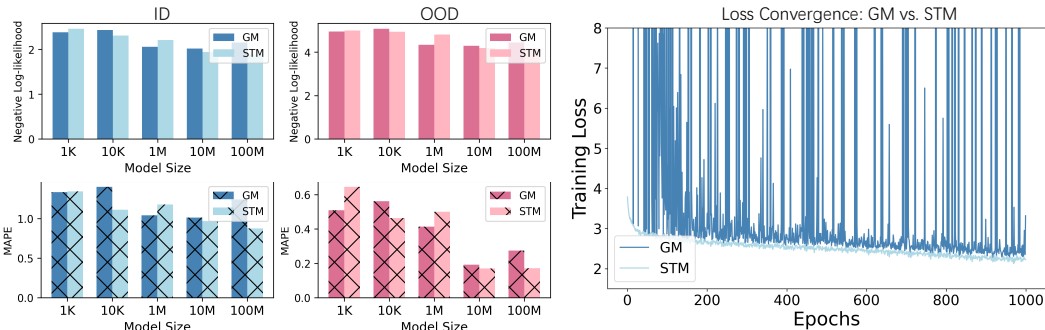

Figure 9: **Gaussian mixture (GM) distribution vs. Student-t mixture (STM) distribution.** *Left*: GM generally under-performs compared to STM in both ID and OOD test data, particularly as model size increases. *Right*: GM exhibits unstable loss convergence during training, while STM achieves smoother convergence. These findings underscore the prevalence of long-tail distributions in real-world time series data and highlights the crucial role of Student-t mixtures in effectively modeling these distributions.

to predict the distribution parameters: degrees of freedom, location, scale, and mixture weights. To ensure positive values for the scale and df, we apply a soft-plus transformation. The mixture weights are constrained within the probability simplex using a soft-max function. To avoid undefined variance for low df values, we impose a lower bound of 2 on the df parameter.

Additionally, we conducted an experiment to compare the performance of Gaussian mixture distribution and Student-t mixture distribution when used as prior distributions to approximate real-world time series. Figure 9 shows that Student-t mixtures generally outperform Gaussian mixtures, particularly as model size increases. Furthermore, we observe unstable convergence when using Gaussian mixture distribution as the prediction head. These findings highlight the prevalence of long-tail distributions in real-world time series data and emphasize the critical role of Student-t mixtures in effectively modeling such distributions.

**Encoder-only Transformer.** The encoder-only Transformer has proven effective for time series tasks (Nie et al., 2023). In input time series, the future horizon is replaced by learnable mask tokens combined with position information. After passing through bidirectional attention blocks, the future representations are derived from these mask tokens and mapped to the parameters of a mixture distribution. Through sampling from the mixture distribution, we can get predictions for future patches.

**Decoder-only Transformer.** The key difference in the decoder-only Transformer is its unidirectional attention. In the input sequence, no masking is needed. The future horizon is predicted based on the preceding token. In other words, given a sequence of input patches, the model is optimized to predict the next patch as a function of all past patches. Similar to LLMs this can be done in parallel over the entire context window, and automatically enables the model to predict the future after having seen varying number of input patches.

## C IMPLEMENTATION DETAILS

### C.1 TRAINING SETUP

Based on the constructed datasets of different sizes, we trained a series of encoder-only Transformers and decoder-only Transformers with increasing data sizes and model parameters. To improve batch processing efficiency and handle varying sequence lengths, we employ sequence packing (Raffel et al., 2020), which reduces padding requirements. Moreover, we train models with forecasting capabilities over varying context and prediction lengths. Rather than defining a fixed context and prediction length, we sample 15% - 50% lengths as forecast horizon and the remaining as context horizon, for a given time series. As most probabilistic forecasting models, we minimize the negative log-likelihood of the predicted patch with respect to the ground truth. The loss convergence

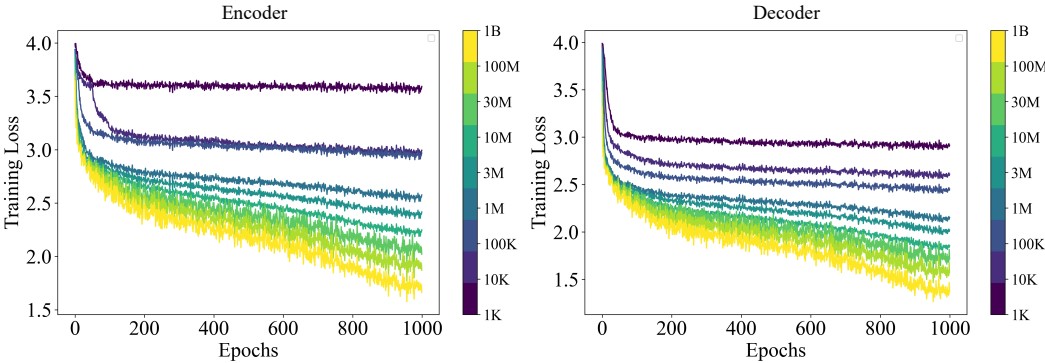

Figure 10: **Convergence Process of Training Loss.** This figure show the training loss convergence for encoder-only (left) and decoder-only (right) models of various sizes, ranging from 1K to 1B parameters. Larger models demonstrate faster convergence and lower final training loss, highlighting their improved capacity for optimization.

process of encoder-only models and decoder-only models is shown in Figure 10. As the number of parameters increases, both encoder-only and decoder-only models tend to converge faster to a lower training loss.

## C.2 EVALUATION SETUP

We primarily used the LSF (Wu et al., 2023) and Monash (Godahewa et al., 2021) datasets to evaluate out-of-distribution generalization ability. For the LSF dataset, we only used the test set and performed evaluations in a non-overlapping rolling window fashion, with the stride equal to the prediction length. For the Monash dataset, considering its large size, we set only one window per sample for evaluation. Table 3 details the evaluation setup per dataset.

Following the statistical method in (Qiu et al., 2024), we compute several statistical characteristics of the datasets used in this study. Table 4 presents the mean statistical features of the pre-training, LSF, and Monash datasets. SNR refers to the signal-to-noise ratio, which quantifies the proportion of meaningful signal relative to random noise in a time series. Shifting captures changes in the probability distribution of a time series over time, with values closer to 1 indicating more severe distributional shifts. Stationarity describes a property of a time series where the mean remains constant over time, and the variance is finite and stable across observations. Transition reflects the regular and identifiable fixed features within a time series, such as trends, periodicity, or other predictable patterns. Overall, our pre-training data and OOD test data both show a high quality and stationarity.

Table 3: Out-of-distribution Evaluation Datasets.

| Dataset | Domain | Frequency | # Prediction Length | # Samples. |
|---|---|---|---|---|
| ETTh1 | Energy | H | 192 | 175 |
| ETTh2 | Energy | H | 192 | 819 |
| ETTm1 | Energy | H | 192 | 819 |
| ETTm2 | Energy | H | 192 | 819 |
| Electricity | Energy | H | 192 | 7062 |
| Weather | Climate | H | 192 | 1029 |
| M4 Hourly | Finance | H | 48 | 414 |
| M4 Daily | Finance | D | 14 | 4227 |
| M4 Weekly | Finance | W | 13 | 359 |

| Table 3 continued from previous page | | | | |
|---|---|---|---|---|
| **Dataset** | **Domain** | **Frequency** | **# Prediction Length** | **# Samples** |
| M4 Monthly | Finance | M | 18 | 48000 |
| M4 Quarterly | Finance | Q | 8 | 24000 |
| M1 Monthly | Finance | M | 18 | 617 |
| M3 Monthly | Finance | M | 18 | 1428 |
| M3 Other | Finance | Q | 8 | 174 |
| NN5 Daily | Finance | D | 56 | 111 |
| NN5 Weekly | Finance | W | 8 | 111 |
| Tourism Monthly | Finance | M | 24 | 366 |
| Tourism Quarterly | Finance | Q | 8 | 427 |
| CIF 2016 | Finance | M | 6 | 15 |
| Traffic Hourly | Transport | H | 168 | 862 |
| Rideshare | Transport | H | 168 | 2304 |
| Saugeen | Climate | D | 30 | 1 |
| Sunspot | Climate | D | 30 | 1 |
| Temperature Rain | Climate | D | 30 | 32072 |
| Vehicle Trips | Transport | D | 30 | 329 |
| Weather | Climate | D | 30 | 3010 |
| Car Parts | Sales | M | 12 | 2674 |
| FRED MD | Finance | M | 12 | 107 |
| Pedestrian Counts | Transport | H | 12 | 66 |
| Hospital | Health | M | 12 | 767 |
| Covid Deaths | Health | D | 30 | 266 |
| KDD Cup 2018 | Energy | H | 168 | 270 |
| Bitcoin | Finance | D | 30 | 18 |
| Us Births | Health | D | 30 | 1 |

Table 4: **Mean Statistic Characteristics.**

| Dataset | SNR | Shifting | Stationarity | Transition |
|---|---|---|---|---|
| Pre-training | 25.47 | 0.33 | 0.47 | 0.03 |
| LSF | 19.78 | 0.15 | 0.50 | 0.01 |
| Monash | 19.84 | 0.34 | 0.38 | 0.09 |

## C.3 EVALUATION METRICS

**Mean Absolute Percentage Error (MAPE).** MAPE is a commonly used metric to evaluate the accuracy of a forecasting model by expressing the error in percentage terms, making it unit-free and easily interpretable. It measures the average magnitude of the absolute percentage error between the predicted and actual values, relative to the actual values. For a uni-variate time series, the error is defined as:

$$e_j^{(i)} = y_j^{(i)} - \hat{y}_j^{(i)}$$

where $y_j^{(i)}$ and $\hat{y}_j^{(i)}$ are the target and predicted values of the $i$-th time series and $j$-th time step, respectively. The MAPE of the $i - th$ time series is then calculated as:

$$\text{MAPE} = \frac{100}{H} \sum_{j=t+1}^{t+H} \frac{|e_j^{(i)}|}{|y_j^{(i)}|}$$

This formula calculates the percentage error at each time step, averaging it over the forecast horizon $H$. MAPE is intuitive and interpretable in percentage terms. However, MAPE can be highly sensitive to small actual values because it divides by $y_j^{(i)}$, leading to extremely large errors when actual values approach zero.

**Symmetric Mean Absolute Percentage Error (SMAPE).** SMAPE addresses the shortcoming of MAPE by using a symmetric formula that normalizes both the actual and predicted values. The SMAPE of the $i$-th time series is defined to be

$$\text{SMAPE} = \frac{200}{H} \sum_{j=t+1}^{t+H} \frac{|e_j^{(i)}|}{|y_j^{(i)}| + |\hat{y}_j^{(i)}|}$$

The SMAPE ensures that errors are balanced between over- and under-prediction by dividing by the sum of the absolute actual and predicted values. This normalization helps avoid extreme errors when actual values are small, but it can also produce undefined results when both actual and predicted values are zero.

**Mean Absolute Scaled Error (MASE).** MASE is a scale-independent metric used to evaluate forecasting accuracy by comparing the absolute error of a forecast to the average error of a naive forecasting method, such as a one-step-ahead random walk. MASE provides a robust way to assess model performance across different datasets with varying scales, addressing the limitations of metrics like MAPE that are sensitive to the magnitude of actual values. For a uni-variate time series, the MASE of the $i$-th time series is calculated as:

$$\text{MASE} = \frac{\frac{1}{H} \sum_{j=t+1}^{t+H} |e_j^{(i)}|}{\frac{1}{T-1} \sum_{k=2}^{T} |y_k^{(i)} - y_{k-1}^{(i)}|}$$

where $|y_k^{(i)} - y_{k-1}^{(i)}|$ represents the difference between consecutive values in the naive forecast. MASE is particularly useful because it accounts for the variability and scale of the time series, making it comparable across datasets. Unlike MAPE, MASE does not produce extreme values for small or zero actual values, making it more stable and reliable for time series evaluation.

**Continuous Ranked Probability Score (CRPS).** Before we can introduce the CRPS, we need to introduce the weighted quantile loss Park et al. (2022), which is a metric normalized over the test set. We first define the $\alpha$-quantile loss, also known as the pinball loss at quantile level $\alpha$, to be:

$$\Lambda_\alpha(q, y) = (\alpha - 1_{y<q})(y - q)$$

The weighted quantile loss is then the normalized sum of quantile losses,

$$\text{wQL}[\alpha] = 2 \frac{\sum_{(i,j)\in\Omega} \Lambda_\alpha(\hat{q}_j^{(i)}(\alpha), y_j^{(i)})}{\sum_{(i,j)\in\Omega} |y_j^{(i)}|}$$

where $\Omega = \{(i,j) \in \mathbb{Z}^2 : 1 \leq i \leq n, \tau_i + 1 \leq j \leq T_i\}$.

The CRPS is a proper scoring rule Matheson & Winkler (1976), meaning that it is minimized when the predictive distribution is equal to the distribution from which the data is drawn.

$$\text{CRPS} = \int_0^1 2\Lambda_\alpha(F^{-1}(\alpha), y) \, d\alpha$$

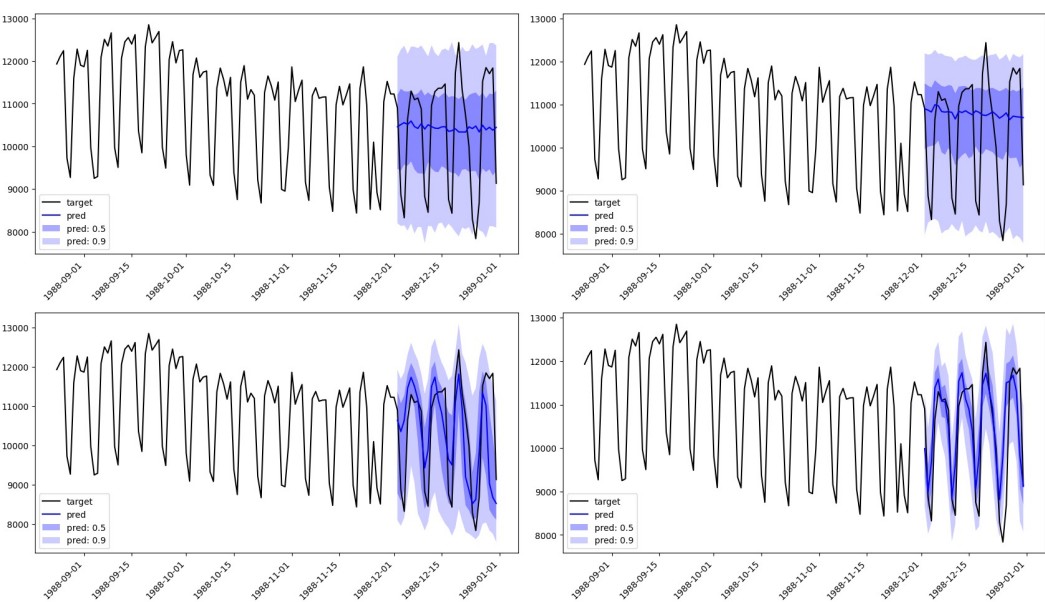

Figure 11: **Emergent Behaviors in us birth Dataset.** The sub-figures, arranged from left to right and top to bottom, show prediction results for models of sizes 10K, 1M, 10M, and 100M. Models with sizes 10K and 1M are unable to capture the periodic pattern, whereas models with sizes 10M and 100M accurately reflect it.

However, we are unable to evaluate this quantity since we generally are not able to compute the integral in closed form and only have access to a finite number of quantile predictions. The approximation of the CRPS is an average of the weighted quantile loss over $K$ quantiles, and thus is also known as the mean weighted quantile loss.

$$\text{CRPS} \approx \frac{1}{K} \sum_{k=1}^{K} \text{wQL}[\alpha_k]$$

# D    ADDITIONAL RESULTS

## D.1    EMERGENT BEHAVIORS

Figure 8 presents case studies on "emergent abilities" of TSFMs for OOD predictions. As model size increases, one would expect the MAPE to follow a smooth power-law decline, indicating continuous improvement. However, the plots reveal distinct phenomena, where performance changes abruptly rather than gradually. Figures 11 - 13 visually show the transition from the small model prediction results to the large model prediction results. We see that the 10K and 1M models are unable to capture the historical seasonal patterns to make accurate predictions; however, when the model size reaches 10M, it can make accurate predictions, far better than the small model. This behavior is akin to the "emergent behaviors" described in (Wei et al., 2022), where certain capabilities absent in smaller models only appear in larger ones. The emergence of improved performance at certain model sizes could indicate the development of higher-order patterns or representations within the model. These emergent abilities might stem from the model's capacity to recognize and generalize complex temporal patterns that were previously inaccessible at smaller scales. Such abilities is evident particularly in zero-shot OOD prediction, where the model must extrapolate from training data to entirely unseen scenarios.

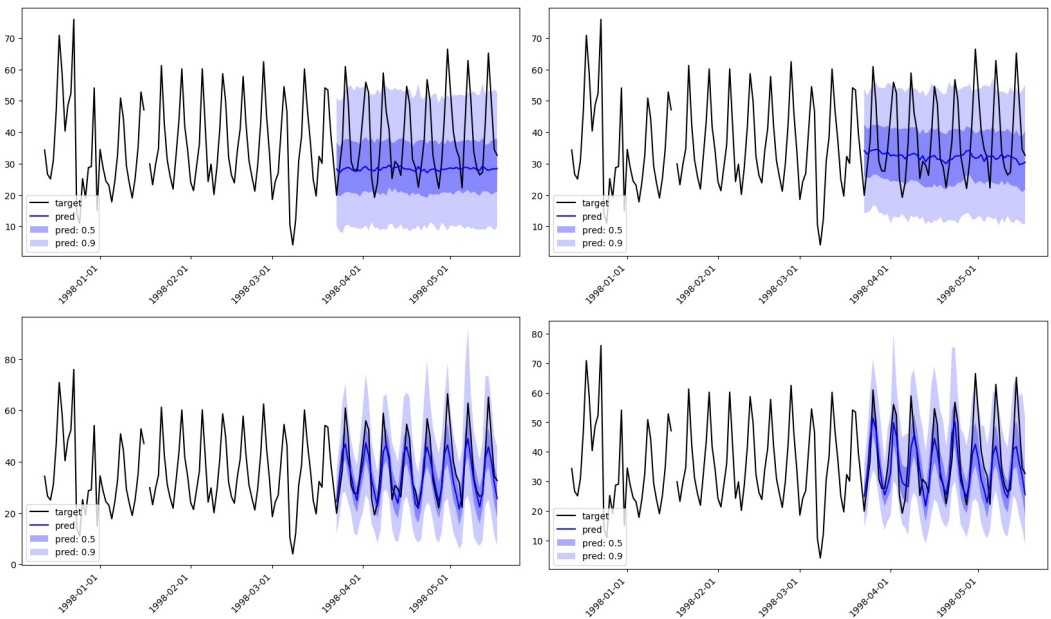

Figure 12: **Emergent Behaviors in nn5 daily Dataset.** From left to right and top to bottom, the sub-figures illustrate the prediction outcomes of models with sizes 10K, 1M, 10M, and 100M, respectively. While the smaller models (10K and 1M) fail to recognize the periodic pattern in the data, the larger models (10M and 100M) effectively capture and reflect the periodic structure.

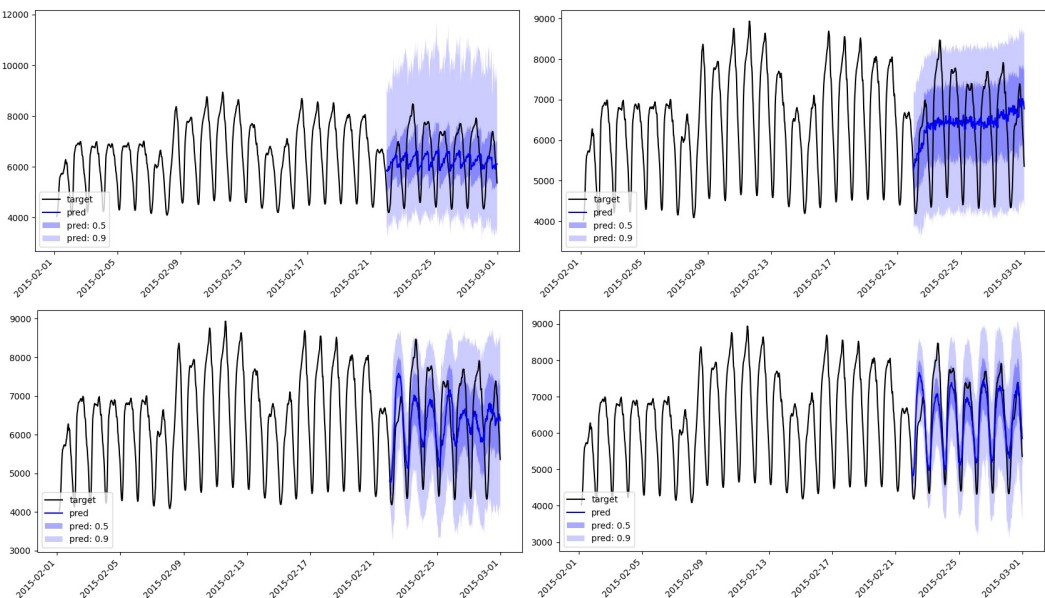

Figure 13: **Emergent Behaviors in australian electricity Dataset.** From left to right and top to bottom, the sub-figures display the prediction results of models with sizes 10K, 1M, 10M, and 100M. The models with sizes 10K and 1M fail to accurately capture the periodic pattern, while those with sizes 10M and 100M successfully do so.

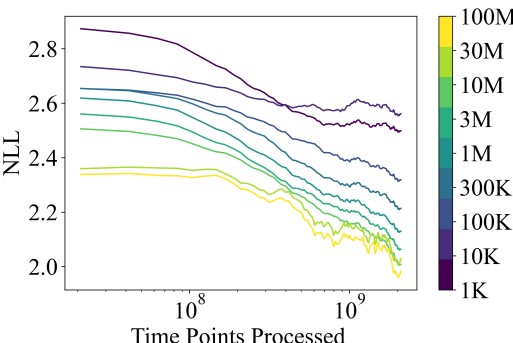

Figure 14: The figure illustrates the validation NLL during training for encoder-only models of different sizes. Larger models tend to achieve lower NLL with processing the same time points.

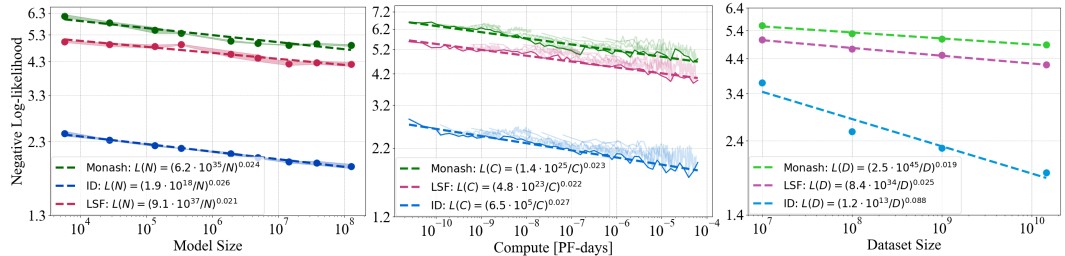

Figure 15: **Scaling Pattern of NLL.** This figure illustrates scaling laws for NLL in relation to model size, compute, and dataset size. The blue lines represent ID performance, while the red and green lines show OOD performance on LSF subset and Monash subset.

## D.2 SAMPLE EFFICIENCY

Figure 14 illustrates the evaluation results on the ID data for encoder-only models of different sizes during training. The key observation is that larger models achieve better performance with fewer training steps compared to smaller models, demonstrating higher sample efficiency. This is important in scenarios where data are limited, as larger models can achieve superior performance without needing to process as much data. An explanatory intuition is that, when the number of parameters is large, even a small change in parameters can cause a large change in outputs. From a theoretical perspective, the Neural Tangent Kernel (Jacot et al., 2018) framework provides insights into why this occurs. As neural networks become infinitely wide, their behavior approximates that of a kernel method, enabling them to effectively minimize the data distribution's generalization error. This allows large models to generalize well with fewer samples, further supporting their observed sample efficiency in empirical settings. Moreover, in NLP (Kaplan et al., 2020) and CV (Zhai et al., 2022) fields, larger models have demonstrated higher sample efficiency than smaller models.

## D.3 SCALING PATTERN DEPENDS ON DATA DISTRIBUTIONS

In Figures 15 - 19, we present the scaling laws evaluated on the Monash subset, along with results from the LSF subset and the ID test data. By comparing the green line, representing the Monash dataset, and the red line, representing the LSF dataset, we observe a relatively consistent offset and similar slopes across various metrics. This suggests that, when transferring the model from the training data distribution to other OOD distributions, there is a predictable decrease in performance that varies by the target dataset. However, the model's gains from scaling—whether by increasing model size, compute, or dataset size—follow a fixed proportional relationship. This indicates that while OOD performance degrades, scaling the model still yields a consistent improvement ratio across different distributions, albeit with varying levels of absolute performance. Moreover, in the future, we can further analyze the reasons why transfer cost arises and establish a law to predict the

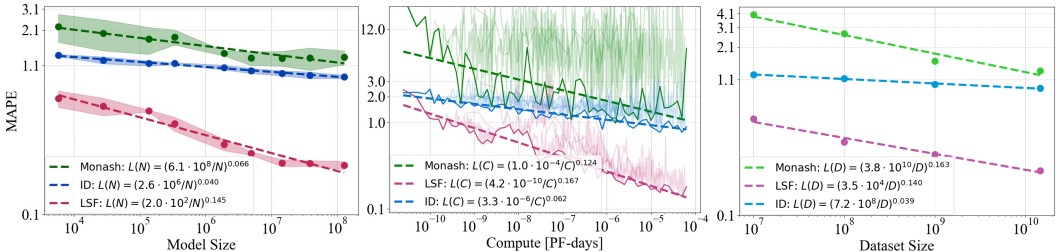

Figure 16: **Scaling Pattern of MAPE.** This figure presents scaling laws for MAPE as functions of model size, compute, and dataset size. ID performance is illustrated by the blue lines, while OOD performance is depicted in red for the LSF subset and green for the Monash subset.

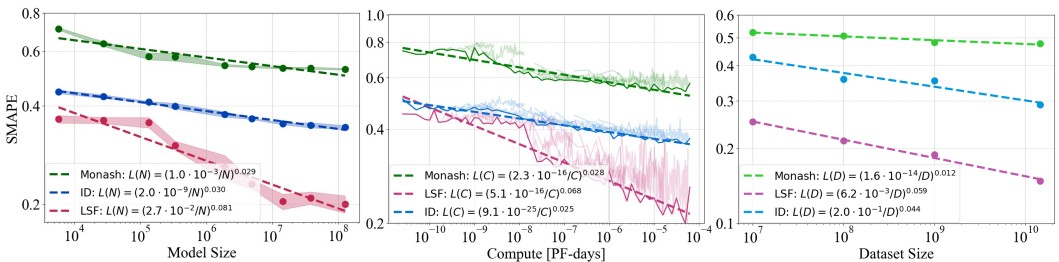

Figure 17: **Scaling Pattern of SMAPE.** This figure depicts the scaling laws for SMAPE with respect to model size, computational resources, and dataset size. ID performance is marked by blue lines, while red and green lines correspond to OOD performance on the LSF and Monash subsets, respectively.

transfer performance loss in population-risk based theoretical analysis (Yang et al., 2021) through Wasserstein measurement.

### D.4 SCALING PATTERN DEPENDS ON PERFORMANCE METRICS.

We investigate the scaling behaviors of five common performance metrics: NLL, MAPE, SMAPE, MASE, and CRPS, (as shown in Figures 15 - 19). All metrics exhibit a decreasing trend following an approximate power-law; however, each metric demonstrates distinct scaling characteristics, reflected in their varying power-law exponents. Previous work by (Ghorbani et al., 2021) establishes a relationship between large language models' log-likelihood loss and the BLEU score in translation tasks. Similarly, future research could explore a transformation between log-likelihood loss and time series forecasting metrics, offering a means to predict forecasting performance from training loss.

## E FURTHER DISCUSSION

**Scaling Laws for Multivariate Time Series Forecasting.** Extending our findings to multivariate time series forecasting is promising but presents some challenges. First, large and diverse multivariate datasets are limited, unlike the more standardized uni-variate time series datasets. This scarcity makes it difficult to conduct experiments that scale both model size and data volume in a controlled manner. Second, establishing scaling laws for multivariate time series require analyzing the impact of variable count and correlation strength, increasing the experimental complexity. Third, multivariate foundation models lag behind uni-variate ones, facing challenges in designing architectures that accommodate variable counts and capture complex inter-variable dependencies.

**Pre-training Data Mixing Strategy.** It's significant to investigate the impact of data mixing strategies on model performance, as different strategies can introduce performance bias or even degrade model performance. Time series pre-training datasets, often comprising data from dozens to hundreds of sources, create exponentially large possibilities for combinations. Instead of exhaustive

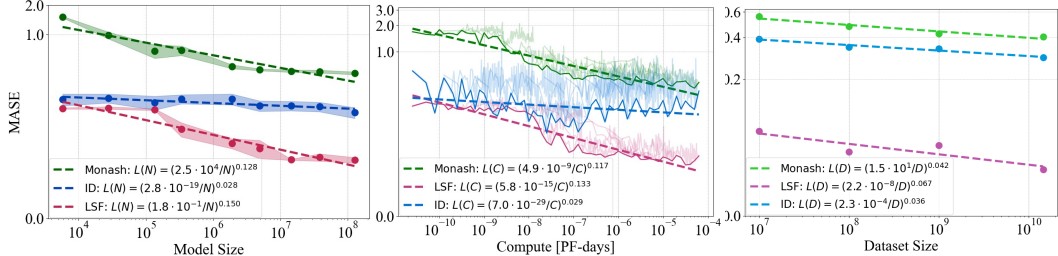

Figure 18: **Scaling Pattern of MASE.** The figure shows how MASE scales with model size, compute, and dataset size. ID performance is indicated by blue lines, whereas OOD performance is highlighted in red and green for the LSF and Monash subsets.

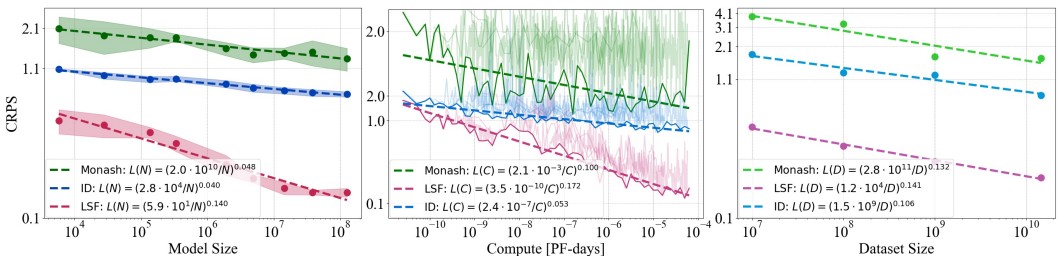

Figure 19: **Scaling Pattern of CRPS.** The figure illustrates the scaling laws for CRPS in relation to model size, compute, and dataset size. Blue lines represent ID performance, while red and green lines depict OOD performance on the LSF and Monash subsets, respectively.

empirical research, a more practical approach is to establish an extrapolable analytical framework that balances experimental complexity (Hashimoto, 2021; Ye et al., 2024). Additionally, deeper analysis of time series dataset characteristics is essential. This would provide insights into the nature of time series data and facilitate the development of better data organization formats, moving beyond simple combination strategies.

**Impact of Model Architecture on Scalability.** Our research comparing the scalability of encoder-only and decoder-only Transformers can serve as the groundwork for future exploration of other architectures. CNN-based models (Wu et al., 2022), RNN-based models (Gu et al., 2021), and hybrid models (Lieber et al., 2024) each demonstrate unique strengths in sequence modeling. Investigating their scalability within the context of time series modeling is a promising direction for future research, and we hope our findings will inspire further work in this area.

**Impact of Specific Module on Scalability.** Our case studies on the scalability of Chronos and Moirai demonstrate that certain module designs in these models may hinder their scalability in OOD settings. Key factors, such as the time series embedding approach, attention mechanism, positional encoding, and forecasting strategy, all impact scalability to varying degrees. Among these factors, the embedding approach is particularly crucial, as it directly determines the input patterns fed into the model backbone and influences the operations the model can learn.

**Theoretical Framework for Scaling Laws.** Our scaling laws, similar to most scaling laws, are empirical findings. We believe a theoretical understanding of the learning dynamics that form the laws can provide a more solid justification. A potential perspective is understanding the optimization process through dynamics modeling (Bordelon et al., 2024). Specifically, based on the dynamical mean field theory, we can derive the infinite limit statistical description of the scaling behavior of training factors. By analyzing the response function in the infinite limit, which gauges the correlation between training factors and test error, we can obtain insights into how the scaling laws manifest.

**Correlation Estimation for Metrics.** Time series forecasting involves a diverse set of metrics, each measuring prediction error from a unique perspective. However, due to the lack of clear and definitive correlations between NLL loss and these metrics, it's difficult to understand model predic-

tion behavior directly from NLL loss. A potential perspective is to develop a prediction framework that bridges these metrics by empirically analyzing their relationships and dependencies (Isik et al., 2024). Such a framework would provide insights into how different metrics reflect model performance under varying conditions, enabling a deeper understanding of prediction behavior.

