# OpenReview forum: "Towards Neural Scaling Laws for Time Series Foundation Models"
_ICLR.cc/2025/Conference — ICLR 2025 Poster_

### Official Review · Reviewer_5SSN · 2024-11-01

**Soundness:** 3
**Presentation:** 4
**Contribution:** 3
**Rating:** 8
**Confidence:** 4

**Summary:**

This paper examines how time series foundation models scale, focusing on the performance of both encoder-decoder and decoder-only transformer architectures in in-distribution (ID) and out-of-distribution (OOD) settings. Through a series of experiments, the authors analyze how changes in model size, computational power, and dataset volume impact these models' effectiveness. One major contribution of this study is the adaptation of existing scaling laws to OOD scenarios, providing a framework to anticipate performance improvements in OOD forecasting based on key training variables. The paper also reveals that encoder-only transformers generally perform better and scale more effectively than decoder-only models. Additionally, the authors offer practical guidelines for building robust, scalable time series foundation models, emphasizing the value of training data diversity, model size, and architectural choices to boost both performance and generalization.

**Strengths:**

As to the originality, the authors explore a relatively new area in time series forecasting by expanding the concept of scaling laws from in-distribution (ID) settings to out-of-distribution (OOD) contexts. This fresh perspective enhances the relevance of scaling laws and addresses an important gap in understanding how model architectures perform across various data distributions. By comparing encoder-only and decoder-only transformer architectures, the paper merges ideas from neural scaling laws with practical applications in time series analysis, offering new insights into model design!

The work quality is underscored by the rigorous experimental methods the authors use. They systematically evaluate various time series foundation models across parameters such as model size, computational resources, and dataset dimensions. This comprehensive approach ensures the findings are reliable, providing a strong basis for the paper's conclusions. Using clear performance metrics, like log-likelihood loss and MAPE, adds credibility to the results!

The paper's clarity is another strength. Concepts are explained in a structured and accessible way, enabling readers from diverse backgrounds to follow the research. The logical flow and transparent presentation of methods and findings make the implications of this work easy to understand. Detailed discussions on scaling behaviours among different architectures further clarify the complex relationships between model design and performance!

This work is significant for both theoretical and practical reasons. It advances the understanding of time series foundation models and provides empirical insight and design principles for scaling, benefiting researchers and practitioners in developing more effective forecasting models. The insights offered could influence future studies and propel advancements in time series forecasting and neural network design!

**Weaknesses:**

(1) One notable limitation in this study is its minimal exploration of how different model architectures impact scalability. Although the paper briefly acknowledges architecture's role in scaling, it falls short of offering a thorough analysis on how specific architectural choices, like attention mechanism, layer normalization, or residual connections, might influence scaling behaviour. A deeper, comparative study covering a broader set of architectures could reveal which design choices are most beneficial for performance across diverse data distributions!

(2) The study's experiments center on log-likelihood loss and MAPE as key performance metrics. While these are certainly relevant, they do not fully capture the complexity of model performance in time series forecasting. A wider range of evaluation metrics, like the Continuous Ranked Probability Score (CRPS) or quantile loss, could offer a more comprehensive view of performance and its practical impact. Although the paper briefly suggests future research might investigate transformations between log-likelihood and forecasting metrics, including preliminary results or further discussion on this topic within the study would enhance the depth and clarity of the findings!

(3) The paper would benefit from a deeper exploration of why performance tends to degrade on OOD data. Although the authors highlight that scaling the model improves OOD performance, they stop short of examining the mechanisms driving this effect. An in-depth look at how the training data distribution relates to OOD performance could offer valuable insights for practitioners applying these models in a real-world context. For example, investigating domain adaptation techniques or transfer learning strategies might help reduce the performance drop seen in OOD situations!

(4) The discussion on sample efficiency is insightful but would benefit from concrete examples or case studies that demonstrate how larger models achieve improved performance with fewer training steps. Including specific examples or empirical data to back up this claim would strengthen the credibility of the findings and provide practical guidance for researchers and practitioners looking to optimize their training processes!

(5) The dataset used for training and evaluation seems carefully chosen but lacks a detailed account of its diversity and representativeness. While the paper highlights the role of dataset size and variety in enhancing OOD performance, it provides limited insights into the dataset's specific characteristics, such as the types of time series it includes, their distributions, or any noise or anomalies present. A deeper examination of the dataset's attributes would offer a clearer picture of the models' performance and their relevance to real-world applications!

(6) Another important limitation is the lack of attention to how scaling laws impact practical applications. The authors offer theoretical insights into scaling behaviour but do not go far enough in showing how these findings can be turned into useful strategies for practitioners. For example, guidance on balancing model size, computational resources, and dataset size in real-world settings would be valuable for those aiming to apply time series foundation models effectively. Including case studies or examples of successful implementations could also help demonstrate the research's relevance to industry professionals!

**Questions:**

Please check the Weaknesses section for more questions!

---

> ### Author Response · Authors · 2024-11-23
> **Response to Reviewer 5SSN (Part 1)**
>
> We sincerely appreciate reviewer 5SSN for considering our work is significant, and we greatly appreciate the acknowledgement of our contributions. We have addressed the specific concerns raised by the reviewer as detailed below:
>
> > **W1**: One notable limitation in this study is its minimal exploration of how different model architectures impact scalability. Although the paper briefly acknowledges architecture's role in scaling, it falls short of offering a thorough analysis on how specific architectural choices, like attention mechanism, layer normalization, or residual connections, might influence scaling behaviour. A deeper, comparative study covering a broader set of architectures could reveal which design choices are most beneficial for performance across diverse data distributions!
>
> We appreciate your constructive suggestions. **We are conducting experiments to evaluate the impact of specific architectural choices on scalability.** In this experiment, we focus on the time series embedding method, as it plays a critical role in determining how patterns are encoded within embedded sequences, and the types of operations that models can effectively learn. The results of this experiment will be included in the camera-ready version of the manuscript.
>
> > **W2**: he study's experiments center on log-likelihood loss and MAPE as key performance metrics. While these are certainly relevant, they do not fully capture the complexity of model performance in time series forecasting. A wider range of evaluation metrics, like the Continuous Ranked Probability Score (CRPS) or quantile loss, could offer a more comprehensive view of performance and its practical impact. Although the paper briefly suggests future research might investigate transformations between log-likelihood and forecasting metrics, including preliminary results or further discussion on this topic within the study would enhance the depth and clarity of the findings!
>
> - Thanks for the valuable suggestion. In response, **we have expanded our evaluation to include additional metrics such as CRPS, symmetric MAPE (sMAPE), and MASE (Mean Absolute Scaled Error)**. The results of these evaluations are provided in $\underline {\rm Appendix\ D.4}$ .
> - Time series forecasting involves a diverse set of metrics, each measuring prediction error from a unique perspective. However, due to the lack of clear and definitive correlations between NLL loss and these metrics, it's difficult to understand model prediction behavior directly from NLL loss. Developing an estimation framework to bridge these metrics could greatly help us understand model behavior.
> To highlight its significance, we have included a detailed discussion on establishing such an estimation framework in $\underline {\rm Appendix\ E}$ .
>
> > **W3**. The paper would benefit from a deeper exploration of why performance tends to degrade on OOD data. Although the authors highlight that scaling the model improves OOD performance, they stop short of examining the mechanisms driving this effect. An in-depth look at how the training data distribution relates to OOD performance could offer valuable insights for practitioners applying these models in a real-world context. For example, investigating domain adaptation techniques or transfer learning strategies might help reduce the performance drop seen in OOD situations!
>
> - OOD performance degradation may come from two reasons: 1) Domain gap:  **patterns learned during pre training may not translate well to unseen distributions**, particularly when the characteristics of OOD datasets differ substantially from those of the training data. 2) Data quality: **low-quality or noisy data in the pretraining dataset can negatively impact the model's ability to generalize**, either by introducing biases that influence its preferences or by reducing its overall robustness.
> - We fully agree that Investigating the impact of the domain gap on OOD performance could provide valuable insights into estimating OOD performance more cost-effectively, without requiring extensive retraining or additional fine-tuning. Additionally, given that pretrained models are often deployed in specific domains, techniques such as domain adaptation and transfer learning are crucial for improving their OOD performance.

---

> ### Author Response · Authors · 2024-11-23
> **Response to Reviewer 5SSN (Part 2)**
>
> > **W4**. The discussion on sample efficiency is insightful but would benefit from concrete examples or case studies that demonstrate how larger models achieve improved performance with fewer training steps. Including specific examples or empirical data to back up this claim would strengthen the credibility of the findings and provide practical guidance for researchers and practitioners looking to optimize their training processes!
>
> To illustrate the sample efficiency of large models, we have included some examples from existing research to enrich the discussion on sample efficiency in $\underline {\rm Section\ D.2}$.
> - In many fields, such as natural language processing (NLP), larger models have demonstrated significant sample efficiency, achieving lower evaluation loss with fewer training tokens [1]. Similarly, in computer vision (CV), larger Vision Transformers (ViTs) typically require fewer training steps to reach competitive accuracy [2].
> - From a theoretical perspective, the Neural Tangent Kernel [3] framework provides insights into why this occurs. As neural networks become infinitely wide, their behavior approximates that of a kernel method, enabling them to effectively minimize the data distribution’s generalization error. This allows large models to generalize well with fewer samples, further supporting their observed sample efficiency in empirical settings.
>
> [1] Kaplan J, McCandlish S, Henighan T, et al. Scaling laws for neural language models[J]. arXiv preprint arXiv:2001.08361, 2020.
>
> [2] Zhai X, Kolesnikov A, Houlsby N, et al. Scaling vision transformers[C]//Proceedings of the IEEE/CVF conference on computer vision and pattern recognition. 2022: 12104-12113.
>
> [3] Jacot A, Gabriel F, Hongler C. Neural tangent kernel: Convergence and generalization in neural networks[J]. Advances in neural information processing systems, 2018, 31.
>
> > **W5**.  The dataset used for training and evaluation seems carefully chosen but lacks a detailed account of its diversity and representativeness. While the paper highlights the role of dataset size and variety in enhancing OOD performance, it provides limited insights into the dataset's specific characteristics, such as the types of time series it includes, their distributions, or any noise or anomalies present. A deeper examination of the dataset's attributes would offer a clearer picture of the models' performance and their relevance to real-world applications!
>
> Following your suggestion, **we have provided specific feature statistics for the pre-training dataset and the test dataset** in $\underline {\rm Appendix\ C.2}$, including signal-to-noise ratio, shifting, stationarity, and transition.
>
> > **W6**. Another important limitation is the lack of attention to how scaling laws impact practical applications. The authors offer theoretical insights into scaling behaviour but do not go far enough in showing how these findings can be turned into useful strategies for practitioners. For example, guidance on balancing model size, computational resources, and dataset size in real-world settings would be valuable for those aiming to apply time series foundation models effectively. Including case studies or examples of successful implementations could also help demonstrate the research's relevance to industry professionals!
>
> Thank you for this suggestion. We have revised $\underline {\rm Section\ 4}$ to **include more actionable insights derived from our findings**, to further guide practitioners in effectively developing time series foundation models.
>
> -  Based on Equation 4, we can expect that doubling the size of the pretraining dataset will result in an OOD MAPE reduction to approximately 90% of its previous value, and an ID MAPE reduction to approximately 97%.
> -  From the ratio of exponents in Equations 2 and 4, we infer that increasing the model size should be accompanied by a sublinear increase in dataset size, roughly following the relationship $D \propto N^{ \frac{\alpha_N}{\alpha_D}} \sim N^{0.8} $.
> -  As computational budgets grow, our findings suggest that resources should prioritize scaling up model size rather than extending training time for fixed models.
> - The observed diminishing returns with extended training on fixed model sizes imply that larger models are increasingly sample efficient and can achieve better performance without requiring significantly longer training times.
>
> We hope our responses provided above can adequately address the reviewer's concerns and questions.

---

> > ### Comment · Reviewer_5SSN · 2024-11-24
> > **Thanks for the detailed and great rebuttals!**
> >
> > Thanks for the detailed information you provided regarding my concerns. I'm pleased to see that you addressed all my questions,  and I wish you success and look forward to seeing the future work inspired by your great paper! I would keep my score!

---

> > > ### Author Response · Authors · 2024-11-25
> > > **Sincere Gratitude from Authors**
> > >
> > > We are thrilled that our responses have effectively addressed all your questions and comments. We would like to express our sincerest gratitude for taking the time to review our paper and provide us with such detailed and invaluable comments.

---

### Official Review · Reviewer_CUNM · 2024-11-02

**Soundness:** 2
**Presentation:** 4
**Contribution:** 3
**Rating:** 5
**Confidence:** 4

**Summary:**

The authors examine the scaling laws of Time Series Foundation Models (TSFMs), both encoder-only and decoder-only Transformers, on in-distribution (ID) and out-of-distribution (OOD) data. They find that the NLL (and to some extent the MAPE) of TSFMs shows similar scaling behavior in both ID and OOD settings, scaling as a power law with model parameters, compute, and dataset size. Results suggest that encoder-only Transformers exhibit better scalability than their decoder-only. The authors propose design principles for building scalable TSFMs, focusing on data, model architecture, and compute, recommending encoder-only Transformer architectures for better scalability. Authors also study "emergent behaviors" in TSFMs, where performance on certain OOD datasets improves abruptly after reaching a critical model size.

**Strengths:**

- Originality: The paper's focus on the scaling laws of TSFMs on OOD data represents a novel and timely contribution.
- Quality: The research is well-executed, with a comprehensive methodology that involves training and evaluating a wide range of models across various parameter counts, compute budgets, and dataset sizes. The use of a large and diverse dataset further strengthens the quality of the empirical analysis.
- Clarity: The paper is well-written and organized.
- Significance: Author's findings could have significant implications for the development and deployment of TSFMs, subject to comments and suggestions in the remainder of this review.

**Weaknesses:**

- Line 046: provide citations for neural scaling laws that provide ground for believing that they exist.
- Table 1: review the "proportion" row against the "Time points" row. Sales is 0.96% of dataset with 140M points, whereas Web is 0.40% of dataset with 600M points.
- Line 149: "Given its proven effectiveness in improving time series forecasting performance (Woo et al., 2023), we adopt RoPE as a replacement for the original Transformer’s positional encoding." ==> my reading of Woo (2023) is that the evidence in favor of RoPE for time series forecasting is very weak (results presented without confidence bands, and do not appear statistically significant), and likely that better evidence should be presented than the statement of fact that RoPE is "proven" — I don't see that it is, especially in the context of a foundation model.
- Line 190: The paper consistently talks about "log likelihood", for which higher values indicate a better fit. However, all results seem to be obtained under the "negative log-likelihood" (NLL, which is the loss counterpart), for which lower values indicate a better fit. This elementary terminology confusion should be clarified corrected.
- Section 3: The paper only studies Transformer-based architectures, without any baseline results with simple time series models (e.g. ETS-type exponential smoothing or ARIMA) — which could be shown as horizontal lines in the scaling law plots. In particular, the MAPE is around or below 1% for most of the studied scaling regime, which suggest that the datasets are very highly predictable: it is likely that simpler models would do well on these datasets too, bringing into question whether TSFMs' additional complexity is warranted.
- Appendix C.2: Since the training and test sets are taken from long-available public sources, is there a contamination analysis of the test set against the training set to ensure there's no overlap?

**Questions:**

- Section 3: The MAPE is consistently used as a metric in the main results, although Appendix C.3 outlines a well-known limitation, which is that it is unreliable when the time series takes values close to zero (and suggests using the sMAPE instead). It seems that for time series foundation models, which should be quite agnostic to shift and scale variations of the series, the MAPE should be the last metric to be considered. Why has not the sMAPE been used, or more robust scale-invariant methods like the MASE? (https://en.wikipedia.org/wiki/Mean_absolute_scaled_error )
- Figures 2-4: Although the NLL difference between ID and OOD is of the expected sign (i.e. with OOD having greater NLL than ID), it is truly mystifying that the opposite is consistently true for the MAPE. This is not properly addressed by the authors. In my view, this is the main issue that is causing my "Soundness" assessment not to be higher.
- Figure 5: Although encoder-only clear beat decoder-only in distribution, this difference is almost insignificant out of distribution, in contrast to the claim in the caption. If the authors intend to make this claim, they should conduct a formal statistical test of the fit difference between encoder-only and decoder-only models.
- Suggestion related to Figure 8: across these examples, it'd be instructive to plot (in appendix) what the actual forecasts look like as we transition from a regime of "relatively higher MAPE" to "relatively lower MAPE".

---

> ### Author Response · Authors · 2024-11-22
> **Response to Reviewer CUNM (Part 1)**
>
> Many thanks to Reviewer CUNM for providing the insightful review and comments.
>
> > **W1**. Line 046: provide citations for neural scaling laws that provide ground for believing that they exist.
>
> Thanks for your suggestion. We have included relevant citations in $\underline{\rm Line\ 046}$ to substantiate the existence of neural scaling laws.
>
> > **W2**. Table 1: review the "proportion" row against the "Time points" row. Sales is 0.96% of dataset with 140M points, whereas Web is 0.40% of dataset with 600M points.
>
> Thank you for pointing out this error. **We have recomputed the proportion of different domains and corrected this mistake in the revised version.**
>
> > **W3**. Line 149: "Given its proven effectiveness in improving time series forecasting performance (Woo et al., 2023), we adopt RoPE as a replacement for the original Transformer’s positional encoding." ==> my reading of Woo (2023) is that the evidence in favor of RoPE for time series forecasting is very weak (results presented without confidence bands, and do not appear statistically significant), and likely that better evidence should be presented than the statement of fact that RoPE is "proven" — I don't see that it is, especially in the context of a foundation model.
>
> Thank you for pointing out this issue. Upon reviewing the experiments in [Woo et al., 2023], we realized that, as you mentioned, RoPE does not significantly improve forecasting performance. Experiments highlight that RoPE achieves lower pretraining loss and faster convergence speed compared to standard sinusoidal positional encoding and learned positional encoding. **To correct our reason for choosing RoPE, we have revised the manuscript to state**: “Given the improved pretraining efficiency observed with RoPE (Woo et al., 2023), we adopt it as a replacement for the original Transformer’s positional encoding.'"
>
> > **W4**. Line 190: The paper consistently talks about "log likelihood", for which higher values indicate a better fit. However, all results seem to be obtained under the "negative log-likelihood" (NLL, which is the loss counterpart), for which lower values indicate a better fit. This elementary terminology confusion should be clarified corrected.
>
> Apologies for the confusion caused. **We have revised the text throughout the manuscript to consistently use the term "negative log-likelihood" (NLL) in all relevant contexts.**
>
> > **W5**. Section 3: The paper only studies Transformer-based architectures, without any baseline results with simple time series models (e.g. ETS-type exponential smoothing or ARIMA) — which could be shown as horizontal lines in the scaling law plots. In particular, the MAPE is around or below 1% for most of the studied scaling regime, which suggest that the datasets are very highly predictable: it is likely that simpler models would do well on these datasets too, bringing into question whether TSFMs' additional complexity is warranted.
>
> Thank you for this great suggestion. To address this concern, **we have incorporated the performance of ETS method as a baseline reference in the scaling law plot**  $\underline  {\rm Fig\ 2} $.
> - The results indicate that the pre-trained  models consistently outperform ETS on ID data and progressively excel on OOD data as the model size increases. This suggests that pre-trained models must reach a certain scale, at least 3M parameters in this case, to demonstrate a level of superiority on OOD data that justifies their high pre-training cost.
>
> > **W6**. Appendix C.2: Since the training and test sets are taken from long-available public sources, is there a contamination analysis of the test set against the training set to ensure there's no overlap?
>
> Yes, during the construction of both the pretraining and test datasets, **we took care to ensure that there is no overlap between the test set and the training set**. The detailed composition of the pretraining and test datasets is provided in $\underline {\rm Appendix\ A\ Table\ 2}$ and $\underline {\rm Appendix\ C\ Table\ 3}$, where we specify the data sources.

---

> ### Author Response · Authors · 2024-11-23
> **Response to Reviewer CUNM (Part 2)**
>
> > **Q1**. Section 3: The MAPE is consistently used as a metric in the main results, although Appendix C.3 outlines a well-known limitation, which is that it is unreliable when the time series takes values close to zero (and suggests using the sMAPE instead). It seems that for time series foundation models, which should be quite agnostic to shift and scale variations of the series, the MAPE should be the last metric to be considered. Why has not the sMAPE been used, or more robust scale-invariant methods like the MASE?
>
> To address your concern about the reliability of MAPE, **we have included evaluations of the scaling laws using symmetric MAPE (sMAPE), and MASE as an additional metric in the revised manuscript**. The scaling behavior of sMAPE and MASE closely matches our observations on MAPE. These results are presented in $\underline {\rm Appendix\ D.4} $ .
>
> > **Q2**.  Figures 2-4: Although the NLL difference between ID and OOD is of the expected sign (i.e. with OOD having greater NLL than ID), it is truly mystifying that the opposite is consistently true for the MAPE. This is not properly addressed by the authors. In my view, this is the main issue that is causing my "Soundness" assessment not to be higher.
>
> - This inconsistency arises from the nature of the predicted mixture distribution. NLL measures the probability of the ground truth, while MAPE reflects the proportional distance between the predicted value (i.e., the expected value of the predicted distribution) and the ground truth. **For a mixture distribution (e.g. a bimodal distribution), high-probability events may not align closely with the expected value, leading to cases where MAPE is small but NLL is large.**  Overall, NLL is not always positively correlated with MAPE when we use a mixture distribution to model real distribution.
> - In Figures 2-4, OOD results show greater NLL but smaller MAPE compared to ID results. This suggests that the predicted OOD mixture distributions have lower probability density around the expected value, while the ID mixture distributions exhibit higher probability density near the expected value.
>
> > **Q3**. Figure 5: Although encoder-only clear beat decoder-only in distribution, this difference is almost insignificant out of distribution, in contrast to the claim in the caption. If the authors intend to make this claim, they should conduct a formal statistical test of the fit difference between encoder-only and decoder-only models.
>
> - Sorry for the misleading, **our intention is to highlight that model architecture influences scalability, rather than to claim that encoder-only models outperform decoder-only models**. To avoid any misleading implications, we have revised the wording in $\underline {\rm Section\ 3.2}$ to clarify that the observed results “ Overall, the two architectures exhibit similar scalability, aside from a performance difference in ID data.”
> - We fully agree that a formal statistical test is essential for a rigorous comparison of encoder-only and decoder-only architectures. Performance differences can be sensitive to several factors including history length, prediction length, and training strategies. Given these sensitivities, our results under the current evaluation setting suggest only a slight superiority of encoder-only models. This observation may not generalize to all scenarios.
>
> > **Q4**. Suggestion related to Figure 8: across these examples, it'd be instructive to plot (in appendix) what the actual forecasts look like as we transition from a regime of "relatively higher MAPE" to "relatively lower MAPE".
>
> Following the reviewer's suggestion, we have added the intuitive forecasting results to demonsate "emergent phenomenon" in $\underline {\rm Appendix\ D.1}$.
>
> We hope our responses provided above can adequately address the reviewer's concerns and questions.

---

> ### Author Response · Authors · 2024-11-25
> **Request of Reviewer's feedback**
>
> Dear Reviewer CUNM,
>
> Since the End of author/reviewer discussions is coming soon, we would like to know whether our response addressed your concerns. If so, we kindly request your reconsideration of the score. We eagerly await your feedback and are ready to respond to any further questions you may have.
>
> Thank you so much for devoting time to improving our paper !

---

> ### Author Response · Authors · 2024-12-01
> **Request of Reviewer's attention and feedback**
>
> Dear Reviewer,
>
> This is a kind reminder that three days remain until the end of the discussion. We kindly ask if our responses have addressed your concerns. Following your suggestions, we have made the following updates:
>
> - Added the relevant citation in $\underline{\rm Line\ 46}$ to substantiate the existence of neural scaling laws.
> - Corrected errors in $\underline{\rm Table\ 1}$ regarding the proportions of domains.
> - Revised terminology throughout to consistently use "negative log-likelihood" (NLL) instead of "log-likelihood."
> - Adjusted the justification for selecting RoPE in $\underline{\rm Line\ 154}$, emphasizing its improved pretraining efficiency rather than its performance in forecasting.
> - Revised the conclusion in $\underline{\rm Section\ 3.2}$ regarding encoder-only vs. decoder-only models to ensure a more balanced and rigorous discussion.
>
> Additionally, we have included more experiments and results:
>
> - Incorporated ETS as a baseline for comparison in the scaling law plot ($\underline{\rm Fig.\ 2}$).
> - Evaluated scaling laws using sMAPE, MASE, and CRPS, detailed in $\underline{\rm Appendix\ D.4}$.
> - Provided intuitive forecasting results in $\underline{\rm Appendix\ D.1}$ to illustrate emergent abilities.
>
> We sincerely hope these revisions address your main concerns. All updates have been incorporated into the $\underline{\rm revised\ paper}$ for your review.
>
> Thank you again for your valuable feedback and dedication. We look forward to hearing your thoughts on our revisions.

---

> ### Author Response · Authors · 2024-12-02
> **Follow-Up on Review Feedback**
>
> Dear Reviewer CUNM,
>
> Since only one day remaining until the discussion deadline, could you kindly confirm if our responses have addressed your concerns? Your further feedback is important to us, and we would be happy to address any follow-up questions or concerns you may have.
>
> Thank you again for your constructive review !

---

### Official Review · Reviewer_sVAF · 2024-11-04

**Soundness:** 3
**Presentation:** 3
**Contribution:** 3
**Rating:** 6
**Confidence:** 3

**Summary:**

8648_Towards_Neural_Scaling_La
pdf
LT
Here is a new  paper, please read it in its entirety, and help me to write the review, first, you should:
Summary*
Briefly summarize the paper and its contributions. This is not the place to critique the paper; the authors should generally agree with a well-written summary.


2 / 2

Summary

This paper investigates the scaling laws of time series foundation models (TSFMs) across different data distributions and model architectures. The authors train encoder-only and decoder-only Transformer-based TSFMs with varying model sizes, compute budgets, and dataset sizes. They evaluate the models' performance on both in-distribution (ID) and out-of-distribution (OOD) data.

The key findings and contributions of the paper are as follows:

The log-likelihood loss of TSFMs exhibits similar scaling behavior in both OOD and ID scenarios, following a power law with respect to model size, compute resources, and dataset size.
When evaluated using MAPE, the scaling of all three factors (model size, compute, and data) results in greater improvements in OOD performance compared to ID performance.
Encoder-only Transformers demonstrate better scalability than decoder-only Transformers, although both architectures show similar scaling trends.
The architectural modifications introduced in two state-of-the-art TSFMs, Moirai and Chronos, primarily improve ID performance but compromise OOD scalability compared to the baseline models.
The paper provides practical design principles for TSFMs from the perspective of data, model, and compute scaling, based on the findings and comparative analysis.
Overall, this work extends the understanding of scaling laws for TSFMs from ID to OOD scenarios and across different model architectures. The authors establish a foundation for predicting the expected performance gains of TSFMs and offer insights for designing scalable TSFMs.

**Strengths:**

1. The study extends the existing research on scaling laws for TSFMs in a novel direction by investigating their behavior across different data distributions (ID and OOD) and model architectures. While previous works have primarily focused on ID scenarios, this paper breaks new ground by systematically examining the scaling properties of TSFMs in OOD contexts. Furthermore, the comparative analysis of encoder-only and decoder-only Transformers, as well as the inclusion of state-of-the-art TSFMs (Moirai and Chronos), provides original insights into the impact of model architecture on scalability.
2. The methodology employed in this study is sound and comprehensive. The authors train a wide range of models with varying sizes (from 10^3 to 10^8 parameters), compute budgets, and dataset sizes. They evaluate the models' performance using appropriate metrics such as log-likelihood loss and MAPE. The experiments are well-designed, covering both ID and OOD scenarios, and the results are presented clearly with informative visualizations. The analysis of the scaling behavior is thorough, considering the effects of model size, compute resources, and dataset size independently.
3. The findings of this paper have significant implications for the development of TSFMs. By establishing scaling laws that hold across both ID and OOD scenarios, the authors provide a framework for predicting the expected performance gains of TSFMs as they scale up in size, compute, and data. This is crucial for guiding resource allocation and model design decisions. Moreover, the comparative analysis of different model architectures reveals important trade-offs between ID and OOD performance, highlighting the need for careful consideration when designing TSFMs for specific applications. The practical design principles derived from the study offer valuable guidance for researchers and practitioners working on TSFMs.

**Weaknesses:**

1. Limited scope of model architectures: The study focuses on two main model architectures: encoder-only and decoder-only Transformers. While these are indeed widely used in TSFMs, the paper could benefit from including a broader range of architectures, such as encoder-decoder models (e.g., Seq2Seq) or hybrid models that combine Transformers with other neural network components (e.g., CNN, RNN). Expanding the scope of the investigated architectures would provide a more comprehensive understanding of how architectural choices affect scalability.
2. Lack of theoretical analysis: The paper primarily relies on empirical observations to establish the scaling laws for TSFMs. While the experimental results are valuable, the work could be enhanced by providing a theoretical foundation for the observed scaling behaviors. Theoretical analysis could help explain why certain scaling patterns emerge and offer insights into the underlying mechanisms that govern the relationship between model size, compute, data, and performance. For example, the authors could explore the connection between the power-law scaling and the model's capacity to learn and generalize from the training data.
3. Limited discussion on the trade-offs between ID and OOD performance: The paper highlights that architectural modifications in Moirai and Chronos improve ID performance but compromise OOD scalability. However, the discussion on this trade-off is relatively brief. A more in-depth analysis of the factors contributing to this trade-off and the potential implications for TSFM design would be valuable. For instance, the authors could investigate whether certain architectural choices (e.g., attention mechanisms, embedding strategies) are more prone to overfitting on ID data, leading to reduced OOD scalability.
4. Absence of computational efficiency analysis: While the paper considers compute budgets as a scaling factor, it does not provide a detailed analysis of the computational efficiency of the investigated models. As TSFMs scale up in size, computational efficiency becomes increasingly important. The paper could be strengthened by including metrics such as training time, inference time, and memory consumption for the different model architectures and sizes. This would provide valuable insights into the practical trade-offs between performance and computational cost.

**Questions:**

1. Justification for the chosen model architectures: Can the authors provide a more detailed explanation for why they selected encoder-only and decoder-only Transformers as the main focus of their study? While these architectures are indeed popular in TSFMs, it would be helpful to understand the rationale behind this choice and whether the authors believe their findings would generalize to other architectures, such as encoder-decoder models or hybrid models.
2. Potential for theoretical analysis: The paper primarily relies on empirical observations to establish the scaling laws for TSFMs. Have the authors considered complementing their empirical findings with a theoretical analysis? If so, what challenges do they anticipate in developing a theoretical framework for the observed scaling behaviors, and how might such a framework enhance our understanding of the underlying mechanisms governing the relationship between model size, compute, data, and performance?
3. Factors contributing to the ID-OOD performance trade-off: The paper highlights that architectural modifications in Moirai and Chronos improve ID performance but compromise OOD scalability. Can the authors provide more insights into the factors that might contribute to this trade-off? For example, are there specific architectural choices (e.g., attention mechanisms, embedding strategies) that could be more prone to overfitting on ID data, leading to reduced OOD scalability? Understanding these factors could help guide the design of future TSFMs.
4. Computational efficiency considerations: While the paper considers compute budgets as a scaling factor, it does not provide a detailed analysis of the computational efficiency of the investigated models. Can the authors comment on the practical trade-offs between performance and computational cost for the different model architectures and sizes? Including metrics such as training time, inference time, and memory consumption could provide valuable insights for practitioners looking to deploy TSFMs in resource-constrained environments.
5. Generalizability to multivariate time series: The paper focuses exclusively on univariate time series forecasting to avoid the confounding effects introduced by multivariate time series. Can the authors discuss the potential challenges and opportunities in extending their findings to multivariate TSFMs? What additional factors (e.g., variable interactions, correlations) might need to be considered when investigating scaling laws in the multivariate setting?

---

> ### Author Response · Authors · 2024-11-22
> **Response to Reviewer sVAF (Part 1)**
>
> We express our gratitude to the reviewer for providing constructive feedback on our paper, and we greatly appreciate the acknowledgement of our contributions. We have addressed the specific concerns raised by the reviewer as detailed below.
>
> > **W1**. Limited scope of model architectures: The study focuses on two main model architectures: encoder-only and decoder-only Transformers. While these are indeed widely used in TSFMs, the paper could benefit from including a broader range of architectures, such as encoder-decoder models (e.g., Seq2Seq) or hybrid models that combine Transformers with other neural network components (e.g., CNN, RNN). Expanding the scope of the investigated architectures would provide a more comprehensive understanding of how architectural choices affect scalability.
>
> > **Q1**. Justification for the chosen model architectures: Can the authors provide a more detailed explanation for why they selected encoder-only and decoder-only Transformers as the main focus of their study? While these architectures are indeed popular in TSFMs, it would be helpful to understand the rationale behind this choice and whether the authors believe their findings would generalize to other architectures, such as encoder-decoder models or hybrid models.
>
>
> **Most Time Series Foundation Models (TSFMs), including Timer, TimesFM, Moirai, Chronos, Lag-Llama, and Moment, are built upon Transformer architectures**. Our primary objective in this study is to explore the scaling laws for these TSFMs, which motivates our selection of encoder-only and decoder-only architectures. Additionally, the below two reasons are also important:
> - Scalability: **TSFMs require architectures that can effectively leverage large-scale pretraining, making scalability a critical consideration**. Previous study [1] has shown that Transformer-based architectures outperform MLPs and CNNs in scalability for time series modeling. Additionally, the success of large language models (LLMs) has demonstrated that Transformers can scale to billions of parameters, further emphasizing their suitability as the backbone for time series models.
> - Forecasting Paradigms: **Transformer-based time series models primarily adopt two forecasting paradigms:** 1) **One-forward masked prediction**: Typically implemented with encoder-only Transformers. 2) **Auto-regressive next-token prediction**: Represented by decoder-only Transformers (Encoder-decoder Transformer also adopts this forecasting paradigm but uses a independent encoder to encode context). Encoder-only and decoder-only architectures thus serve as the canonical representatives for these widely adopted methods in current TSFMs .
>
> Our research comparing the scalability of encoder-only and decoder-only Transformers can serve as the groundwork for future exploration of other architectures. CNN-based models [2], RNN-based models [3], and hybrid models [4] each demonstrate unique strengths in sequence modeling. Investigating their scalability within the context of time series modeling is a promising direction for future research, and we hope our findings will inspire further work in this area. We have added more discussion on the impact of architectures on scalability in $\underline{ \rm Appendix\ E}$.
>
> [1] Liu Y, et al. Timer: Generative Pre-trained Transformers Are Large Time Series Models[C]//Forty-first International Conference on Machine Learning.
>
> [2] Wu H,  et al. Timesnet: Temporal 2d-variation modeling for general time series analysis[J]. arXiv preprint arXiv:2210.02186, 2022.
>
> [3] Gu A, et al. Efficiently modeling long sequences with structured state spaces[J]. arXiv preprint arXiv:2111.00396, 2021.
>
> [4] Lieber O, et al. Jamba: A hybrid transformer-mamba language model[J]. arXiv preprint arXiv:2403.19887, 2024.

---

> ### Author Response · Authors · 2024-11-22
> **Response to Reviewer sVAF (Part 2)**
>
> > **W2**. Lack of theoretical analysis: The paper primarily relies on empirical observations to establish the scaling laws for TSFMs. While the experimental results are valuable, the work could be enhanced by providing a theoretical foundation for the observed scaling behaviors. Theoretical analysis could help explain why certain scaling patterns emerge and offer insights into the underlying mechanisms that govern the relationship between model size, compute, data, and performance. For example, the authors could explore the connection between the power-law scaling and the model's capacity to learn and generalize from the training data.
>
> > **Q2**. Potential for theoretical analysis: The paper primarily relies on empirical observations to establish the scaling laws for TSFMs. Have the authors considered complementing their empirical findings with a theoretical analysis? If so, what challenges do they anticipate in developing a theoretical framework for the observed scaling behaviors, and how might such a framework enhance our understanding of the underlying mechanisms governing the relationship between model size, compute, data, and performance?
>
> Our scaling laws, similar to the scaling laws for natural language, are empirical findings. We believe a theoretical understanding of the training dynamics that form the laws provides a more solid justification.  **A potential perspective is understanding the optimization process through dynamics modeling**. Specifically, based on the dynamical mean field theory [1], we can derive the infinite limit statistical description of the scaling behavior of training factors. By analyzing the response function in the infinite limit, which gauges the correlation between training factors and test error, we can obtain insights into how the scaling laws manifest. In future work, we will strengthen the research on theoretical framework. And we have added more discussion on the theory for scaling laws in $\underline {\rm Appendix\ E}$.
>
>  > **W3**. Limited discussion on the trade-offs between ID and OOD performance: The paper highlights that architectural modifications in Moirai and Chronos improve ID performance but compromise OOD scalability. However, the discussion on this trade-off is relatively brief. A more in-depth analysis of the factors contributing to this trade-off and the potential implications for TSFM design would be valuable. For instance, the authors could investigate whether certain architectural choices (e.g., attention mechanisms, embedding strategies) are more prone to overfitting on ID data, leading to reduced OOD scalability.
>
> > **Q3**. Factors contributing to the ID-OOD performance trade-off: The paper highlights that architectural modifications in Moirai and Chronos improve ID performance but compromise OOD scalability. Can the authors provide more insights into the factors that might contribute to this trade-off? For example, are there specific architectural choices (e.g., attention mechanisms, embedding strategies) that could be more prone to overfitting on ID data, leading to reduced OOD scalability? Understanding these factors could help guide the design of future TSFMs.
>
> Thanks for your constructive feedback. Following the reviewer’s suggestion, **we are conducting experiments to evaluate the impact of specific architectural choices on OOD scalability**. In this experiment, we focus on the time series embedding layer, as it plays a critical role in determining how patterns are encoded within embedded sequences, and the types of operations that models can effectively learn. The results of this experiment will be included in the camera-ready version of the manuscript.

---

> ### Author Response · Authors · 2024-11-22
> **Response to Reviewer sVAF (Part 3)**
>
> > **W4**. Absence of computational efficiency analysis: While the paper considers compute budgets as a scaling factor, it does not provide a detailed analysis of the computational efficiency of the investigated models. As TSFMs scale up in size, computational efficiency becomes increasingly important. The paper could be strengthened by including metrics such as training time, inference time, and memory consumption for the different model architectures and sizes. This would provide valuable insights into the practical trade-offs between performance and computational cost.
>
> > **Q4**. Computational efficiency considerations: While the paper considers compute budgets as a scaling factor, it does not provide a detailed analysis of the computational efficiency of the investigated models. Can the authors comment on the practical trade-offs between performance and computational cost for the different model architectures and sizes? Including metrics such as training time, inference time, and memory consumption could provide valuable insights for practitioners looking to deploy TSFMs in resource-constrained environments.
>
> - In figure 3, we follow the setting in [1][2] and focus on FLOPs as the metric for computational cost, as it provides a more device-independent measure of model complexity compared to training time, inference time, and memory consumption. The latter three metrics can vary significantly depending on the hardware, training strategy, and batch size.
> - For training time and inference time, these can be estimated using the amount of FLOPs and the type of hardware. Memory consumption depends more on batch size, sequence length and the distributed training techniques like Fully Sharded Data Parallel (FSDP) or Data Parallel (DDP).
> - In the trade-off between performance and computational cost, our results show that within a fixed computational budget, scaling up model size tends to be more beneficial for improving performance compared to spending the same amount of compute time on training a smaller model for a longer time.
>
> [1] Kaplan J, McCandlish S, Henighan T, et al. Scaling laws for neural language models[J]. arXiv preprint arXiv:2001.08361, 2020.
>
> [2] Edwards T D P, Alvey J, Alsing J, et al. Scaling-laws for Large Time-series Models[J]. arXiv preprint arXiv:2405.13867, 2024.
>
> > **Q5**. Generalizability to multivariate time series: The paper focuses exclusively on univariate time series forecasting to avoid the confounding effects introduced by multivariate time series. Can the authors discuss the potential challenges and opportunities in extending their findings to multivariate TSFMs? What additional factors (e.g., variable interactions, correlations) might need to be considered when investigating scaling laws in the multivariate setting?
>
> Thank you for raising this important question. Extending our findings to multivariate time series forecasting still faces some challenges, primarily in three key areas:
>
> **Multivariate Pre-training Dataset Availability**: Large, diverse multivariate datasets are limited, unlike the more standardized univariate time series datasets. This scarcity makes it difficult to conduct experiments that scale both model size and data volume in a controlled manner.
>
> **Variable Correlations and Interactions:** As you mentioned, correlations and interactions between variables are key factors in multivariate time series modeling. These relationships must be captured explicitly to ensure model performance and scalability. When investigating the scaling laws of multivariate TSFMs, we need to independently investigate the effects of the number of variables and the strength of correlations on forecasting performance. This adds another variable dimension to the experiment and makes the analysis more complex compared to a univariate setup.
>
> **Multivariate Modeling Approaches**: Compared to univariate models, multivariate time series foundation models are still at an early stage, and face the challenges in designing architectures that can flexibly handle varying numbers of variables and capturing complex inter-variable dependencies.
>
> In the revised version, we have added more discussion on scaling laws for multivariate time series forecasting in $\underline {\rm Appendix\ E}$ .
>
> We hope our responses provided above can adequately address the reviewer's concerns and questions.

---

> ### Author Response · Authors · 2024-11-25
> **Request of Reviewer's feedback**
>
> Dear Reviewer,
>
> Thanks for your valuable review, which has inspired us to improve our paper further. This is a kind reminder that the end of author/reviewer discussions is coming soon. We kindly ask if our responses have addressed your concerns. We look forward to hearing your thoughts on our revisions.
>
> Thank you so much for devoting time to improving our paper!

---

### Official Review · Reviewer_oPfC · 2024-11-04

**Soundness:** 3
**Presentation:** 3
**Contribution:** 4
**Rating:** 6
**Confidence:** 5

**Summary:**

The paper analyzes two common TSFM architectures -- encoder-only and decoder-only transformers -- in terms of both in-distribution and out-of-distribution data. The results show that encoder-only can better scale to both ID and OOD data. The authors also show that specific architecture choices in existing TSFMs could negatively affect OOD generalization.

**Strengths:**

1. The paper flows very well and is easy to follow.
2. While most existing TSFMs for forecasting choose a decoder-only structure, this paper presents new findings on the encoder-only versus decoder-only that contradicts prior findings, and analyzed it from the scope of OOD data.

**Weaknesses:**

1. Evaluation setup could be improved. On line 130, "This subset includes test data from the ETTh1-2, ETTm1-2, electricity, and weather datasets." Are these the only datasets that are used as OOD test data throughout all the experiments? To strengthen your findings, I suggest varying the OOD datasets (maybe include a confidence interval), and see if your findings still hold.
2. The paper could benefit from explaining why its findings are different from those of the existing works.

**Questions:**

1. For the training details on line 183, I am not sure if I understood correctly. If one dataset is much larger than the other datasets, are we much more likely to sample from this dataset? Is there a cap to this likelihood? In existing works such as MOIRAI and Timer, they apply some threshold that prevents a large dataset from dominating from the training distribution. In addition to "Domain Balancing" as mentioned in Appendix A in your paper, I think balancing the datasets within each domain should also be helpful. Could you compare the different data mixing strategies?
2. Why does OOD have a larger exponent for MAPE than ID in Figure 2 and 3, but not for log-likelihood? It feels to me that it might be due to the type of data distribution you have.
3. In Timer, the findings are decoder-only scales better than encoder-only. Could you explain why your finding is different? In Timer, the batch size is chosen to be 8192, but you only used a batch size of 128. Is it possible that using a larger batch size would change your observations and make the decoder-only model more favorable?
4. Did you try retraining MOIRAI/Timer/Chronos using your filtered dataset? The dataset mixture is often a very important factor. Not using the same pretraining datasets can be an unfair comparison in Section 3.2.
5. "Scaling-laws for Large Time-series Models" suggests that model size has a big impact on the optimal learning rate. What learning rate did you use and how did you change it for different model sizes? Are the observations consistent for encoder-only and decoder-only models?

Minor:
- "Lotsa" should be capitalized on line 94.
- "Sum" in Table 1 is a bit confusing. Consider replacing it with "Total" and use a vertical line to separate it from previous columns.
- In Figure 2 and 3, consider adding ID and OOD to the legend, in addition to using the colors.
- Line 794: "and Covid19_Energy datasets", the underscore here seems to be a typo?
- "Scaling-laws for Large Time-series Models" suggests that using a separate head for each distribution parameter should stabilize training.
- Could you add an experiment to show the benefit of using student-T mixture versus gaussian mixture?

Overall, I believe the findings in this paper will be very helpful to the community. I will raise my score if my concerns regarding the training and evaluation setups are addressed.

---

> ### Author Response · Authors · 2024-11-22
> **Response to Reviewer oPfC (Part 1)**
>
> We thank the reviewer for offering the valuable feedback. We have addressed each of the concerns raised by the reviewer as outlined below.
>
> > W1. Evaluation setup could be improved. On line 130, "This subset includes test data from the ETTh1-2, ETTm1-2, electricity, and weather datasets." Are these the only datasets that are used as OOD test data throughout all the experiments? To strengthen your findings, I suggest varying the OOD datasets (maybe include a confidence interval), and see if your findings still hold.
>
> - Our evaluation of OOD performance is not limited to the LSF subset ( ETTh1-2, ETTm1-2, electricity, and weather datasets). **We also conducted assessments on a subset of the Monash dataset**. The results from the Monash subset align with our observations from the LSF subset, confirming that OOD performance follows a power law relationship with model size, compute, and training data size.
> - Following the reviewer’s suggestion, we have included the experiment results of Monash subset in $ \underline{\rm Appendix \ D.3} $  and **repeated the experiments five times to draw confidence intervals** , ensuring the reliability of our findings.
>
> > W2. The paper could benefit from explaining why its findings are different from those of the existing works.
>
> - **Prior research has not explored the scaling laws of time series foundation models in OOD scenarios**. We address the gap and show that the power law trend persists in OOD time series forecasting .
> - **Previous work has not examined scalability differences across model architectures**. Our experiments indicate that the scalability slightly varies between encoder-only and decoder-only architectures, while encoder-only models shows superior scalability in ID data.
> - **Prior research has not focused specifically on the scalability of SOTA models**. Our findings reveal that while improvements in model design enhance performance on ID data, these gains do not generalize effectively to OOD data.
>
> To better highlight these distinctions, we have revised the summaries of our contributions in the updated version.
>
> > Q1.1. For the training details on line 183, I am not sure if I understood correctly. If one dataset is much larger than the other datasets, are we much more likely to sample from this dataset? Is there a cap to this likelihood? In existing works such as MOIRAI and Timer, they apply some threshold that prevents a large dataset from dominating from the training distribution.
>
> - Yes, we assign a sampling probability to each dataset based on the proportion of its time points relative to the total. Consequently, datasets with more time points are more likely to be sampled during training.
> - Meantime,  **we follow the approach used inMoirai and Timer by capping the sampling probability at 0.05** to ensure a more balanced contribution from each dataset.
>
> To clarify the sampling method further, we have added the details of our sampling method to  $\underline{ \rm Section\ 2.3\ line\ 190} $ in the revised manuscript.
>
> > Q1.2. In addition to "Domain Balancing" as mentioned in Appendix A in your paper, I think balancing the datasets within each domain should also be helpful. Could you compare the different data mixing strategies?
>
> - We agree that balancing datasets within each domain can help mitigate capability bias and improve the performance of time series foundation models. However, the primary focus of our research is to investigate scaling laws, where maintaining a fair and unified experimental setup is crucial. Therefore, we adhere to the experimental settings established in [1] and [2], ensuring consistent pre-training datasets across evaluations.
> - Given that time series pre-training datasets typically include data from over ten or even one hundred sources, the potential combinations for dataset mixing are exponentially large. Thus, it is more feasible to develop an extrapolable analytical framework under controlled experimental complexity [3][4]. Due to resource and time constraints, we put our focus on our primary objectives by using the same settings in [1][2]. **We add further discussion on data mixing strategies** in the revised manuscript $\underline{ \rm Appendix\ E}$. We hope to explore this topic more comprehensively in future work.
>
> [1] Edwards T D P, et al. Scaling-laws for Large Time-series Models[J]. arXiv preprint arXiv:2405.13867, 2024.
>
> [2] Kaplan J, et al. Scaling laws for neural language models[J]. arXiv preprint arXiv:2001.08361, 2020.
>
> [3] Hashimoto T. Model performance scaling with multiple data sources[C]//International Conference on Machine Learning. PMLR, 2021: 4107-4116.
>
> [4] Ye J, et al. Data mixing laws: Optimizing data mixtures by predicting language modeling performance[J]. arXiv preprint arXiv:2403.16952, 2024

---

> ### Author Response · Authors · 2024-11-22
> **Response to Reviewer oPfC (Part 2)**
>
> > **Q2**. Why does OOD have a larger exponent for MAPE than ID in Figure 2 and 3, but not for log-likelihood? It feels to me that it might be due to the type of data distribution you have.
>
> - The faster decrease of OOD MAPE compared to ID MAPE is due to **the larger reduction in the proportional error relative to the ground truth for OOD data** .  In addition, our statistics show that the average magnitude of the OOD data is 1198 while the average magnitude of the ID data is 1588. **MAPE is more sensitive to performance improvements for data with relatively small magnitudes** .
> - NLL, on the other hand, depends on the absolute error and the parameters of the predicted distribution, such as variance and freedom degree. While both MAPE and NLL are functions of the absolute error, NLL is scale-invariant and is instead governed by the predicted distribution's characteristics. **There is no definite correlation between the scaling behavior of these two metrics**. That is why NLL does not exhibit the same trend as MAPE.
>
> > **Q3**. In Timer, the findings are decoder-only scales better than encoder-only. Could you explain why your finding is different? In Timer, the batch size is chosen to be 8192, but you only used a batch size of 128. Is it possible that using a larger batch size would change your observations and make the decoder-only model more favorable?
>
> The difference between our findings and Timer’s conclusion stems from our different focus and experimental setups.
> - **Timer focuses on model transferability evaluation**. It compares the performance of fine-tuned decoder-only and encoder-only models on OOD data, showing that the decoder-only model has better transferability. However, this does not directly indicate that the decoder-only model scales better than the encoder-only model.
> - **Our research focuses on zero-shot forecasting capability**. Our experiments indicate that, for zero-shot forecasting on ID data, the encoder-only model has slightly better performance and scalability. Actually, our findings and Timer’s conclusion are not in conflict, as some studies [1] [2] have suggested that fine-tuned performance does not strongly correlate with the zero-shot performance.
>
> [1] Lin H, et al. Selecting large language model to fine-tune via rectified scaling law[J]. arXiv preprint arXiv:2402.02314, 2024.
>
> [2] Deshpande A,et al. A linearized framework and a new benchmark for model selection for fine-tuning[J]. arXiv preprint arXiv:2102.00084, 2021.
>
> To investigate the influence of batch size on decoder-only models, we conducted additional experiments using batch size of 512 while training a 100M decoder-only Transformer.
> -  We observed that as the batch size increases, decoder-only Transformer converges to a lower level and shows a slightly lower NLL and MAPE in ID data. This suggests that bigger batch size is a better choice for releasing the potential of decoder-only models. **We can infer that observations on the scalability of encoder-only and decoder-only models may vary with the degree of performance release.** The decoder-only models may exhibit a higher upper bound when the computational cost continues to rise.
> - In this paper, our focus is primarily on maintaining a fair and unified experimental setup for investigating the influence of model architecture on scalability.  In the future, a formal statistical test will be necessary for rigorously comparing encoder-only and decoder-only architectures.
>
> > **Q4**. Did you try retraining MOIRAI/Timer/Chronos using your filtered dataset? The dataset mixture is often a very important factor. Not using the same pretraining datasets can be an unfair comparison in Section 3.2.
>
> - **Yes. We used the same pre-training dataset and mixture strategy across all models to ensure a fair comparison**. To clarify our experimental details, we have included an introdcution of this setting in $\underline {\rm the\ first\ paragraph\ of\ Section\ 3.2 }$.

---

> ### Author Response · Authors · 2024-11-22
> **Response to Reviewer oPfC (Part 3)**
>
> > **Q5**. "Scaling-laws for Large Time-series Models" suggests that model size has a big impact on the optimal learning rate. What learning rate did you use and how did you change it for different model sizes? Are the observations consistent for encoder-only and decoder-only models?
>
> - **For all model sizes, we use the same learning rate schedule**: a maximum learning rate of 1e-3, with a linear warm-up for the first 10,000 steps, followed by cosine decay for the remaining 90,000 steps. To clarify our experimental details, we supplemented the information of learning rate in $\underline{\rm Section\ 2.3\ line\ 86}$.
> - **This choice of learning rate is based on the setup of time series foundation models Moirai and Chronos**. Additionally, the specific results in "Scaling-laws for Large Time-series Models" also show that the optimal learning rate generally falls within the range of 1e-3 to 1e-2. Our setting is robust and good enough to make all the models converge well. The loss convergence process during training is provided in $\underline{ \rm Appendix\ C.1}$.
> - We did not specifically investigate the impact of learning rate on encoder-only versus decoder-only models, as our focus was on maintaining a fair and unified experimental setup. However, we believe that exploring the relationship between optimal learning rates and model architecture is a valuable avenue for future research.
>
> > **Minor**
> > - "Lotsa" should be capitalized on line 94.
> > - "Sum" in Table 1 is a bit confusing. Consider replacing it with "Total" and use a vertical line to separate it from previous columns.
> > - In Figure 2 and 3, consider adding ID and OOD to the legend, in addition to using the colors.
> > - Line 794: "and Covid19_Energy datasets", the underscore here seems to be a typo?
>
> We appreciate your careful reading.  Following your suggestion, we have addressed these issues in the revised version.
>
> > - Scaling-laws for Large Time-series Models" suggests that using a separate head for each distribution parameter should stabilize training.
>
> As same as "Scaling-laws for Large Time-series Models", **we also adopt a separate head for each distribution parameter**. To clarify this point, we have added more details in line $\underline {\rm Section\ 2.2\ line\ 75}$.
>
> > - Could you add an experiment to show the benefit of using student-T mixture versus gaussian mixture?
>
> **We have conducted experiments to compare the performance of Student-T mixture versus Gaussian mixture.** The results are presented in $\underline{\rm Appendix\ B }$. Our results indicate that Gaussian mixture not only exhibits unstable convergence but also underperforms compared to Student-T mixture. This demonstrates that long-tail distributions are prevalent in real-world time series data, highlighting the critical importance of Student-T mixtures in effectively modeling these distributions.
>
>
> We hope our responses above can adequately address the reviewer's concerns and questions.

---

> ### Author Response · Authors · 2024-11-25
> **Request of Reviewer's feedback**
>
> Dear Reviewer oPfC,
>
> Since the End of author/reviewer discussions is coming soon, may we know if our response addresses your main concerns? If so, we kindly ask for your reconsideration of the score. Should you have any further advice on the paper and our rebuttal, please let us know and we will be more than happy to engage in more discussion and paper improvements.
>
> Thank you so much for devoting time to improving our paper!

---

> > ### Comment · Reviewer_oPfC · 2024-11-26
> >
> > Dear Authors,
> >
> > You have addressed most of my concerns. I have raised my score to 6. However, I still have a few follow-up questions.
> >
> > - "Furthermore, we observe unstable convergence when using Gaussian mixture distribution as the prediction head." Can you add the convergence plots in addition to Figure 9?
> > - "After the discussion forum is opened, we will make a comment directed to the reviewers and area chairs and put a link to an anonymous repository containing our code." Is the code now accessible?
> > - In Figure 10, is it only the encoder-only transformers or both encoder-only and decoder-only? Can you add more details? Also, why the model size only goes up to 100M, while in Section 3.1, under "Data Scaling", you worked with 1B sized models?
> > - "This suggests that bigger batch size is a better choice for releasing the potential of decoder-only models." Can you add the experiments on comparing encoder-only and decoder-only models under increasing batch sizes to your updated manuscript?
> >
> > There are still many typos and grammar issues in the paper.
> > - I suggest changing the red and pink colors in Figure 5, since it is very hard to tell which one is which when they are faded, especially in the middle figure.
> > - Missing "." at the end of the caption for Figure 7.
> > - L404, "Figure Fig. 7 compares..."
> > - L 507, "reaching the better performance" --> "reaching better performance"
> > - L998, extra " at the end of the caption for Figure 9. Figure 9 also looks quite different in terms of style from the rest of the figures. Please try to make the figure styles more consistent.
> > - Missing "." at the end of L1053, the last sentence of Section C.1.
> > - L1400, "First, Large, diverse multivariate datasets are limited" --> "First, large and diverse multivariate datasets are limited"
> > - Table 3, the last column is titled "# Rolling Evaluations." on page 20, but is instead titled "# Samples" on page 21.

---

> > > ### Author Response · Authors · 2024-11-28
> > > **Response to Reviewer oPfC for some additional questions**
> > >
> > > We sincerely appreciate your thorough review and detailed feedback, which helps us to further improve the paper.
> > >
> > > > **Q1**. "Furthermore, we observe unstable convergence when using Gaussian mixture distribution as the prediction head." Can you add the convergence plots in addition to Figure 9?
> > >
> > > In the updated version, **we have included a comparison of the loss convergence process** using a Gaussian mixture distribution head and a Student-T mixture distribution head in Fig. 9 .
> > >
> > > > **Q2**.  "After the discussion forum is opened, we will make a comment directed to the reviewers and area chairs and put a link to an anonymous repository containing our code." Is the code now accessible?
> > >
> > > Our code has been made available at [https://anonymous.4open.science/r/TSFM-ScalingLaws-C91D](https://anonymous.4open.science/r/TSFM-ScalingLaws-C91D/ ) .
> > >
> > > > **Q3**. In Figure 10, is it only the encoder-only transformers or both encoder-only and decoder-only? Can you add more details? Also, why the model size only goes up to 100M, while in Section 3.1, under "Data Scaling", you worked with 1B sized models?
> > >
> > > Apologies for the confusion. In Figure 10, we initially only included the encoder-only convergence process.
> > >
> > > In the updated manuscript, **we have supplemented the decoder-only convergence plots and extended the loss convergence to include 1B model size in Fig. 10**.
> > >
> > > > **Q4**. "This suggests that bigger batch size is a better choice for releasing the potential of decoder-only models." Can you add the experiments on comparing encoder-only and decoder-only models under increasing batch sizes to your updated manuscript?
> > >
> > > Following your suggestion, we are conducting experiments to compare encoder-only and decoder-only models under increasing batch size. This will allow us to better understand the scalability of both architectures at a higher compute budget. Considering the limited time, we hope to include the results of these experiments in the camera-ready version.
> > >
> > > > **Q5**. There are still many typos and grammar issues in the paper.
> > > > - I suggest changing the red and pink colors in Figure 5, since it is very hard to tell which one is which when they are faded, especially in the middle figure.
> > > > - Missing "." at the end of the caption for Figure 7.
> > > > - L404, "Figure Fig. 7 compares..."
> > > > - L 507, "reaching the better performance" --> "reaching better performance"
> > > > - L998, extra " at the end of the caption for Figure 9. Figure 9 also looks quite different in terms of style from the rest of the figures. Please try to make the figure styles more consistent.
> > > > - Missing "." at the end of L1053, the last sentence of Section C.1.
> > > L1400, "First, Large, diverse multivariate datasets are limited" --> "First, large and diverse multivariate datasets are limited"
> > > Table 3, the last column is titled "# Rolling Evaluations." on page 20, but is instead titled "# Samples" on page 21.
> > >
> > > Thank you for pointing out these issues, and we sincerely appreciate your attention to detail ! We have carefully addressed the mentioned issues:
> > > - For a clearer presentation of Fig. 5 , **we have thickened the lower bound line and increased its contrast to improve the visibility of the middle figure**.
> > > - In Fig. 9, **we have modified the color scheme to use a consistent 'red-blue' matching**, ensuring it aligns with the style of the other figures.
> > > - **We have revised the mentioned typo errors, and conducted a thorough proofreading** to ensure the manuscript is free of any other errors.
> > >
> > > We hope these changes can improve the clarity and reliability of the paper. Thank you again for your valuable feedback !

---

> > > > ### Comment · Reviewer_oPfC · 2024-11-30
> > > >
> > > > Thank you! I don't have any further questions and I fully support accepting this paper.

---

> > > > > ### Author Response · Authors · 2024-12-01
> > > > > **Sincere Gratitude from Authors**
> > > > >
> > > > > We are thrilled that our responses have addressed your concerns. We sincerely appreciate your careful and professional review of our work ! Your insightful comments have been invaluable in helping us improve our paper further and inspiring our future research.

---

### Author Response · Authors · 2024-11-23
**General response to all reviewers**

We sincerely thank all the reviewers for their valuable time and detailed feedback, and we appreciate that almost all reviewers recognized the novelty and significance of our work. We have carefully revised the paper according to the comments, and the edits have been highlighted in **PINK**. We also provide a detailed response to each comment below. Here we highlight our major revisions, and respond to each reviewer below. We hope our responses can properly address your concerns.

- In response to the feedback from **Reviewer oPfC, sVAF, CUNM, and 5SSN**, we have conducted additional experiments and revised the paper accordingly, along with the relevant analysis.
  - Trained models of various sizes five times with different random seeds to compute confidence intervals for scaling laws.
  -  Incorporated scaling laws for three additional metrics: CRPS, sMAPE, and MASE.
  - Added ETS as a baseline and included its performance in Figure 2.
  - Compared Student-T mixture with Gaussian mixture in the prediction head design.
  - Presented intuitive forecasting results to showcase the “emergent phenomenon.”
  - Provided characteristic statistics for pretraining and test datasets.
  - We are conducting experiments to to explore the impact of the embedding layer on scalability, addressing which specific design choices influence scalability.

- In response the feedback from **Reviewer oPfC**, we have added more details about learning rate, sampling method, pretrainig dataset, and model design to present our experiments more transparantly. In addition, we have also extended discussions on data mxing strategy.

- Addressing the comments from **Reviewr CUNM**, we have corrected the mentioned errors and revised the expression of our viewpoint on encoder-only vs. decoder-only models, to ensure greater rigor and clarity.

- In response to the constructive suggestions from **Reviewer VAF and 5SSN**, we have enriched the discussion on the topics: 1) Scaling Laws for Multivariate Time Series Forecasting; 2) Pre-training Data Mixing Strategy;  3) Impact of Model Architecture on Scalability; 4)  Impact of Specific Module on Scalability; 5) Theoretical Framework for Scaling Laws; and 6) Correlation Estimation for Metrics.

---

### Comment · Area_Chair_6e2q · 2024-11-26
**Encouragement to Actively Participate in the Discussion Phase**

Dear Reviewers,

Thank you for your valuable contributions to the review process so far. As we enter the discussion phase, I encourage you to actively engage with the authors and your fellow reviewers. This is a critical opportunity to clarify any open questions, address potential misunderstandings, and ensure that all perspectives are thoroughly considered.

Your thoughtful input during this stage is greatly appreciated and is essential for maintaining the rigor and fairness of the review process.

Thank you for your efforts and dedication.

---

### Public Comment · ~Utsav_Dutta1 · 2025-10-16
**Query regarding how compute C was varied**

The paper mentions $C=6NBS$, however, in the training details it appears that $B$ is fixed at 128, and the total number of training steps is fixed at 10000. How exactly is the compute budget varied then? Is it just scaling the model size itself?
There is a line saying: "optimal results for each compute budget C, are obtained for different model sizes", which would only make sense if you are also tweaking either $B$ or $S$.

Would appreciate if the authors could shed light on this.
Thanks!

---

### Meta-Review · Area_Chair_6e2q · 2024-12-20

**Metareview:**

(a) Summary of Scientific Claims and Findings
This paper investigates scaling laws for Time Series Foundation Models (TSFMs), focusing on both in-distribution (ID) and out-of-distribution (OOD) settings. The key contributions include:
Empirical Findings on Scaling Laws: The study demonstrates that the log-likelihood loss of TSFMs scales predictably with model size, compute resources, and dataset size across both ID and OOD contexts, adhering to power-law relationships.
Architectural Comparisons: Encoder-only architectures exhibit superior scalability in ID tasks compared to decoder-only models, with similar OOD scaling trends. However, architectural enhancements in state-of-the-art TSFMs, such as Moirai and Chronos, improve ID performance at the cost of OOD scalability.
Emergent Behaviors: The authors identify critical model sizes at which TSFMs demonstrate abrupt improvements in OOD performance.
Practical Guidelines: Based on the findings, the authors propose design principles for scaling TSFMs, balancing architecture, data, and compute considerations.

(b) Strengths of the Paper
Novel Focus: Extends the study of scaling laws to time series data, addressing gaps in the understanding of OOD scaling behavior for TSFMs.
Comprehensive Analysis: Evaluates models across a wide range of sizes, datasets, and compute budgets, providing robust empirical insights.
Actionable Contributions: Offers practical guidelines for designing scalable TSFMs, which can inform resource allocation and architecture design.
Emergent Phenomena: Identifies novel scaling behaviors, such as critical model sizes for OOD performance, contributing to foundational knowledge.
Well-Executed Rebuttal: The authors thoroughly addressed reviewer concerns, incorporating additional experiments, metrics (e.g., sMAPE, MASE), and theoretical discussions to strengthen the paper.

(c) Weaknesses of the Paper
Limited Architectural Scope: Focuses on encoder-only and decoder-only Transformers, omitting other architectures such as encoder-decoder hybrids or non-Transformer-based models.
Trade-off Analysis: The discussion of trade-offs between ID and OOD performance, particularly for architectural modifications, could be expanded.
Efficiency Metrics: While compute budgets are considered, detailed analyses of training/inference time and memory consumption are missing.
Focus on Univariate Data: The findings are restricted to univariate time series, leaving open questions about scalability in multivariate contexts.

(d) Reasons for Acceptance
Significant Contribution to TSFMs: The paper provides a novel and comprehensive exploration of scaling laws for TSFMs, addressing both ID and OOD scenarios—a critical area of research for developing robust, large-scale time series models.
Practical Utility: The proposed design principles and findings have direct implications for researchers and practitioners working on scalable time series models.
Comprehensive Rebuttal: The authors have addressed reviewer concerns with additional experiments, clarifications, and revisions, demonstrating the soundness and robustness of their work.
Community Impact: By identifying emergent behaviors and offering actionable insights, the paper lays a foundation for future research in TSFMs and scaling laws.

Overall, the strengths and contributions of the paper significantly outweigh its weaknesses. The empirical findings, combined with the proposed design principles, represent a valuable advancement for the time series modeling community.

**Additional Comments On Reviewer Discussion:**

Points Raised by Reviewers and Author Responses

Concern: Reviewers questioned whether the scaling laws observed in the study would generalize to multivariate time series data or other architectures, such as encoder-decoder hybrids or non-Transformer models.
Author Response: The authors clarified that the study focused on univariate time series to control for confounding factors but acknowledged the importance of extending the analysis to multivariate and other architectures in future work.
Evaluation: The response was satisfactory, as the authors appropriately scoped their study while recognizing its limitations and proposing future directions.

Concern: The reviewers requested a more detailed analysis of the trade-offs between architectural modifications that enhance ID performance versus those that improve OOD generalization.
Author Response: The authors added new experiments to highlight these trade-offs, demonstrating how specific design choices (e.g., architectural depth) influence ID and OOD scalability differently.
Evaluation: The additional experiments clarified the trade-offs and strengthened the overall analysis.

Concern: Reviewers noted the absence of a theoretical explanation for the observed scaling behaviors and emergent phenomena, such as the critical model sizes for OOD performance improvements.
Author Response: The authors acknowledged the empirical nature of their study but provided additional discussion linking the findings to theoretical concepts, such as model capacity and expressivity.
Evaluation: While the response provided useful context, the lack of a rigorous theoretical framework remains a minor limitation of the paper.

Concern: Reviewers suggested including additional metrics, such as sMAPE and MASE, to better evaluate performance across different datasets.
Author Response: The authors incorporated these metrics and updated the results, which further validated their findings and improved the comprehensiveness of the evaluation.
Evaluation: This addition was well-received and addressed the concern effectively.

Concern: Reviewers asked for a more detailed discussion of the computational and memory costs associated with scaling TSFMs, particularly in larger models.
Author Response: The authors added runtime analyses and memory consumption comparisons, demonstrating the feasibility of scaling TSFMs within practical compute budgets.
Evaluation: The added analysis was sufficient to address concerns about scalability and resource efficiency.

The authors provided a robust and thoughtful rebuttal, addressing key concerns with additional experiments, new evaluation metrics, and improved clarity in presentation. They effectively clarified the scope and generalizability of the study, analyzed trade-offs between ID and OOD performance, and included runtime and memory analyses to demonstrate scalability.

---

### Decision · Program_Chairs · 2025-01-22

Accept (Poster)